# Unleashing the Potential of Text-attributed Graphs: Automatic Relation Decomposition via Large Language Models

## Abstract

Recent advancements in text-attributed graphs (TAGs) have significantly improved the quality of node features by using the textual modeling capabilities of language models. Despite this success, utilizing text attributes to enhance the predefined graph structure remains largely unexplored. Our extensive analysis reveals that conventional edges on TAGs, treated as a single relation (*e.g.*, hyperlinks) in previous literature, actually encompass mixed semantics (*e.g.*, "advised by" and "participates in"). This simplification hinders the representation learning process of Graph Neural Networks (GNNs) on downstream tasks, even when integrated with advanced node features. In contrast, we discover that decomposing these edges into distinct semantic relations significantly enhances the performance of GNNs. Despite this, manually identifying and labeling of edges to corresponding semantic relations is labor-intensive, often requiring domain expertise. To this end, we introduce **RoSE** (**R**elation-**o**riented **S**emantic **E**dge-decomposition), a novel framework that leverages the capability of Large Language Models (LLMs) to decompose the graph structure by analyzing raw text attributes - in a *fully automated* manner. **RoSE** operates in two stages: (1) identifying meaningful relations using an LLM-based generator and discriminator, and (2) categorizing each edge into corresponding relations by analyzing textual contents associated with connected nodes via an LLM-based decomposer. Extensive experiments demonstrate that our model-agnostic framework significantly enhances node classification performance across various datasets, with accuracy improvements of up to 16%.

## 1 Introduction

Text-attributed graphs (TAGs) (Yang et al., 2021), which combine graph structures with textual data, are frequently used in diverse real-world applications, including fact verification (Zhou et al., 2019; Liu et al., 2019), recommendation systems (Zhu et al., 2021), and social media analysis (Li et al., 2022). In TAGs, texts are incorporated as node descriptions such as paper abstracts in citation networks (McCallum et al., 2000; Sen et al., 2008; Hu et al., 2020a) or web page contents in hyperlink networks (Mernyei & Cangea, 2007; Craven et al., 1998). By leveraging the rich information present in both the graph topology and its associated text attributes, substantial advancements have been achieved in graph representation learning. Among them, numerous studies have been proposed to enhance the node representation quality of TAGs by leveraging features generated from light-weighted pre-trained language models (PLMs) (Yang et al., 2021; Chien et al., 2021; Zhao et al., 2022; Dinh et al., 2023; Duan et al., 2023; Jin et al., 2023a; Chen et al., 2024) such as Sentence-BERT (Reimers & Gurevych, 2019), or by refining raw texts using the general knowledge of Large Language Models (LLMs) (He et al., 2023; Chen et al., 2024).

Despite their success, the potential of utilizing text attributes to enhance the predefined *graph structure* remains largely under-explored. Existing approaches have treated the edges in TAGs as a uniform relation, overlooking the diverse inherent semantics they convey. For instance, in the WebKB dataset (Craven et al., 1998), nodes denote web pages with their textual content as node features while their edges are formed by hyperlinks. Despite the presence of varying semantic meanings such as "node A is advised by node B" or "node A participates in node C", the relationships are bundled as a single relation type ("hyperlinks"), inadvertently entangling their semantic meanings.

Our comprehensive analysis reveals that the downstream task performance of GNNs is hindered by these oversimplified graph structures, even when integrating node features obtained from PLMs. On the other hand, disentangling edges into multiple semantic types — analogous to the knowledge graph format — yields more distinguishable representations and significantly enhances performance. However, such conversion of conventional graphs is extremely labor-intensive, as it requires both the identification of semantic edge types and the classification of numerous edges into their corresponding types.

To address these challenges, we propose **RoSE** (**R**elation-**o**riented **S**emantic **E**dge-decomposition), a novel framework that utilizes LLMs to decompose predefined edges into semantic relations via textual information of nodes in a *fully-automated* manner. Given the graph description and textual content, **RoSE** carefully identifies a concise set of meaningful relation types through the interaction between an LLM-based generator and a discriminator. Subsequently, the LLM-based decomposer disentangles each edge into predefined relation types by analyzing raw textual contents associated with its connected nodes. The versatility of our proposed framework is readily extended to varying architectures, encompassing edge-featured GNNs (Hu et al., 2020c; Shi et al., 2020; Rampášek et al., 2022) and multi-relational GNNs (Schlichtkrull et al., 2018; Wang et al., 2019; Yang et al., 2023). In essence, **RoSE** is a data enhancement method tailored for real-world TAGs that *frequently* lack edge-wise information, in alignment with prior works that leverage the textual reasoning capabilities of LLMs for data augmentation in other domains (Yang et al., 2024; Chen et al., 2023; Dixit et al., 2022; Korenčić et al., 2022).

Our contributions are summarized as follows:

- We reveal that the oversimplified graph structure in TAGs hinders the performance of GNNs on downstream tasks despite the integration of informative node features. On the other hand, mitigation through decomposing graph edges lead to significant enhancements in GNN performance.
- We present **RoSE**, a novel edge decomposition framework that utilizes the general reasoning capability of LLMs. **RoSE** identifies semantic relations through the interaction between an LLM-based generator and discriminator, and categorizes each edge into these relation types by analyzing node textual contents via LLM-based decomposer. All these processes are automated, eliminating the need for extensive human analysis and annotation.
- Extensive evaluations on diverse TAGs and GNN architectures demonstrate the effectiveness of **RoSE** in improving node classification performance. Notably, our framework achieves improvements of up to 16% on the Wisconsin dataset.

## 2 PRELIMINARIES

**Node classification with graph neural networks.** We study a TAG $\mathcal{G} = (\mathcal{V}, \mathcal{E}, \mathcal{T})$, comprising $N$ nodes in $\mathcal{V}$ along with a node-wise text attribute $\mathcal{T} = \{t_i | i \in \mathcal{V}\}$ and $M = |\mathcal{E}|$ undirected edges connecting nodes. Nodes are characterized by a feature matrix $\boldsymbol{X} = [\boldsymbol{x}_1, \boldsymbol{x}_2, ..., \boldsymbol{x}_N]^\mathsf{T} = g_{\boldsymbol{\phi}}(\mathcal{T}) \in \mathbb{R}^{N \times F}$, where their text attributes are encoded using a PLM $g_{\boldsymbol{\phi}}$ which is typically frozen. Edges are described by a binary adjacency matrix $\boldsymbol{A} \in \mathbb{R}^{N \times N}$, with $\boldsymbol{A}[i, j] = 1$ if an edge $(i, j) \in \mathcal{E}$, and $\boldsymbol{A}[i, j] = 0$ otherwise.

Our focus lies on a node classification task using a GNN $f_{\boldsymbol{\theta}}$. The GNN learns representation of each node $i$ by iteratively aggregating representations of its neighbors in the neighborhood set $\mathcal{N}_i$ in the previous layer, formulated as:

$$\boldsymbol{h}_i^{(l+1)} = \psi\big(\boldsymbol{h}_i^{(l)}, \ \texttt{AGG}(\{\boldsymbol{h}_j^{(l)}, \forall j \in \mathcal{N}_i\})\big). \tag{1}$$

Here, AGG denotes an aggregation function and $\psi$ combines the node's prior representation with that of its aggregated neighbors. The initial representation is $\boldsymbol{h}_i^{(0)} = \boldsymbol{x}_i$ for notational simplicity and the overall multi-layered process can be expressed as $f_{\boldsymbol{\theta}}(\boldsymbol{X}, \boldsymbol{A})$. The objective function $\mathcal{L}$ used for training the GNN is defined as the cross-entropy loss between the predicted class probabilities $\boldsymbol{P} = \text{Softmax}(\boldsymbol{Z}) = \text{Softmax}\big(f_{\boldsymbol{\theta}}(\boldsymbol{X}, \boldsymbol{A})\big) \in \mathbb{R}^{N \times K}$ and the ground-truth labels $\boldsymbol{Y} \in \mathbb{R}^{N \times K}$:

$$\mathcal{L}_{\boldsymbol{\theta}} = -\frac{1}{N} \sum_{i \in \mathcal{V}}^{N} \sum_{k=1}^{K} \boldsymbol{Y}_{ik} \log \boldsymbol{P}_{ik}, \tag{2}$$

where $\boldsymbol{Z}$ represents the logit produced by the GNN and $K$ represents the total number of classes.

Table 1: Node classification accuracy (%) on WebKB and IMDB datasets, trained with single and multi-type relations, averaged over 10 runs ($\pm$ SEM). The best performances are represented by **bold**.

| Datasets | | Cornell | Texas | Wisconsin | IMDB |
|---|---|---|---|---|---|
| RGCN | Single Type | $57.60 \pm 1.78$ | $65.88 \pm 1.86$ | $59.22 \pm 1.70$ | $62.96 \pm 0.44$ |
| | **Multi Type** | **$68.80 \pm 1.88$** | **$76.47 \pm 1.82$** | **$83.28 \pm 1.64$** | **$68.66 \pm 0.57$** |
| HAN | Single Type | $56.00 \pm 1.67$ | $68.82 \pm 2.12$ | $58.28 \pm 1.99$ | $63.24 \pm 0.54$ |
| | **Multi Type** | **$60.40 \pm 1.91$** | **$71.37 \pm 2.24$** | **$76.09 \pm 1.88$** | **$68.39 \pm 0.62$** |

**Prompting large language models.** LLMs pre-trained on a vast amount of text corpora have demonstrated remarkable general reasoning capabilities proportional to their number of parameters (Brown et al., 2020; Ouyang et al., 2022; Touvron et al., 2023; Chowdhery et al., 2023). This advancement has led to a new approach to task alignment, allowing for the direct output obtainment from natural language prompts without the need for additional fine-tuning (Kojima et al., 2022; Wei et al., 2022; Liu et al., 2023b). In practice, a natural language text prompt $s$ is concatenated with a given input sequence $q = \{q_i\}_{i=1}^n$ to form a new sequence $\widetilde{q} = \{s\} \cup q$. Subsequently, an LLM $\mathcal{M}$ receives $\widetilde{q}$ as its input and generates an output comprising a sequence of tokens $a = \{a_i\}_{i=1}^m = \mathcal{M}(\widetilde{q})$.

## 3 ANALYSIS: UNCOVERING THE IMPORTANCE OF SEMANTIC EDGE DECOMPOSITION

In this section, we analyze the potential performance improvements of GNNs when applied to TAGs with available semantic edge types. Toward this, we choose three TAG datasets of a small size enough to manually classify the semantic types of edges. First, we perform our analysis on WebKB hyperlink graphs (Cornell, Texas, Wisconsin) (Craven et al., 1998), where nodes represent web pages and edges indicate hyperlinks between nodes. Despite traditionally being treated as single relation graphs, their edges can be mainly categorized into multiple semantic types, such as "participates in", "advises/advised by", "being part of", and "supervised by". To the best of our knowledge, this is the first analysis to broadly create and label relation types in such graphs to verify GNNs' performance in a multi-relational scenario. Additionally, we include the IMDB graph (Fu et al., 2020), which consists of movie nodes with edges reflecting overlaps between movie professionals. In contrast to the WebKB graphs, the edges in the IMDB graph have been consistently regarded as multi-relations (Wang et al., 2019; Yun et al., 2019), differentiated into "actor/actress overlap" and "director overlap". By incorporating this dataset into our analysis, we demonstrate the potential performance degradation when inherent relations are simplified as a single relation.

We evaluate the efficacy of relation labeling under the node classification task, with two multi-relational GNN architectures; namely RGCN (Schlichtkrull et al., 2018) and HAN[1] (Wang et al., 2019). Each is an extension of GCN (Kipf & Welling, 2016) and GAT (Veličković et al., 2017) to multi-relational scenarios, equipped with an edge type-specific neighborhood aggregation scheme (detailed formulation is outlined in Section 4.3). Note that in the case of training with a single relation, RGCN and HAN function similarly to asymmetric GCN and GAT, correspondingly. We train these GNNs in two different approaches: processing edges as a single and multiple types of relation.

As demonstrated in Table 1, decomposing edges into multiple semantic relations leads to significant performance improvements across all datasets and GNN architectures. This enhancement is particularly pronounced in the Wisconsin dataset, where accuracy improvements of 26.56% and 19.37% are achieved for RGCN and HAN, respectively. Furthermore, our analysis reveals that neglecting the entangled semantics in multi-relational benchmark results in suboptimal performance. The benefits of decomposition are also evident at the representation level, showing more distinguishable and clustered node representations, as illustrated in Figure 3 and Figure 4 in Appendix B. Hence, our observation highlights the suboptimality present within the graph structure due to its oversimplification of edges, which can be adequately addressed through the decomposition of edges into distinct semantic relations.

---

[1] Due to the scope of our research on semantic edge decomposition, we do not consider node type-wise aggregation in HAN.

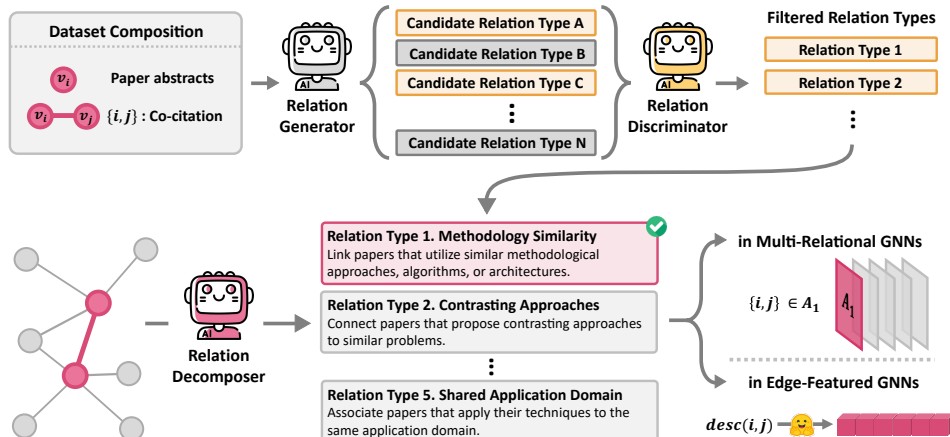

Figure 1: Overall framework of our **RoSE**. **RoSE** automaticaly decomposes graph edges into multiple semantic relations using a two-stage process: (1) identifying candidate relation types via *relation generator-discriminator* interaction and (2) assigning each edge to its appropriate relation type based on its associated nodes' textual attributes analyzed by *relation-decomposer*.

## 4 ROSE: RELATION-ORIENTED SEMANTIC EDGE-DECOMPOSITION

Despite the efficacy of semantic edge decomposition introduced in Section 3, the practical implementation of semantic edge decomposition presents several challenges. To begin with, defining the appropriate semantic relation type is a non-trivial task that often requires domain expertise. Additionally, creating annotations for the numerous edge types is extremely labor-intensive. In turn, this limits the application of fine-tuned PLMs for edge decomposition, as they necessitate the identified list of edge types and the ground-truth edge labels for fine-tuning.

To address this, we present **RoSE**, an innovative framework that leverages the advanced textual reasoning capabilities of LLMs to automate the decomposition of edges into their inherent semantic relations based on their corresponding text attributes. **RoSE** is structured into two main phases: (1) Relation Type Identification (Section 4.1), and (2) Semantic Edge Decomposition (Section 4.2). The edges decomposed by **RoSE** can be seamlessly integrated with conventional GNN architectures in a plug-and-play manner (Section 4.3). This is facilitated either through direct edge type-specific neighborhood aggregation in multi-relational GNNs or by assigning relation types as edge features in edge-featured GNNs. In addition, to enhance efficiency, we introduce an edge sampling strategy that reduces the number of queries required for LLM-based edge type annotation (Section 4.4). It is worth noting that our edge decomposition is accomplished within single-round LLM queries, eliminating the necessity for re-computation or further fine-tuning required by previous LLM-based feature enhancement methods (He et al., 2023; Duan et al., 2023; Chien et al., 2021). The overall framework of **RoSE** is illustrated in Figure 1.

### 4.1 RELATION TYPE IDENTIFICATION

To decompose each edge into underlying semantic relations, it is essential to identify relation types that are: (1) meaningful, capturing the inherent context of predefined edges; (2) feasible, determinable based solely on textual attributes; and (3) distinct, ensuring clarity and avoiding redundancy within the graph.

We use a combination of an LLM-based *relation generator* and *relation discriminator* for this task. The *relation generator* addresses the requirement for meaningfulness by generating a set of plausible candidate relations based on the graph composition. The *relation discriminator* ensures feasibility and distinctiveness by filtering out candidate relation types that exceed the analytical capability of LLMs or exhibit excessive redundancy. The effectiveness of this generator - discriminator framework is outlined in Section 5. We provide detailed information of each component in the following paragraphs. All prompt templates fixed throughout our experiments is specified in Appendix A.

**Relation generator.** To obtain a set of edge types relevant to the given graph, we provide the *relation generator* $\mathcal{M}_g$ with detailed information about the graph in the input prompt $\boldsymbol{s}_g$, which is mathematically formulated as $\mathcal{M}_g(\boldsymbol{s}_g)$. This information includes specifying node's textual attributes along with a corresponding sample (*e.g.*, paper abstracts), predefined rules for node connectivity (*e.g.*, co-citation), and category names (*e.g.*, rule learning). Subsequently, we outline the role of $\mathcal{M}_g$ and specifies the preliminary requirements for identifying meaningful relations within the graph. Based on the provided graph composition and task description, the *relation generator* generates a list of candidate relation types in a zero-shot manner, without any additional fine-tuning.

**Relation discriminator.** To ensure the feasibility and distinctiveness of the generated relation types, we employ a *relation discriminator* $\mathcal{M}_d$. The discriminator $\mathcal{M}_d$ takes the relation types generated by $\mathcal{M}_g$ as input and filters out those that are irrelevant or infeasible to infer given the textual attributes and the analytical capabilities of LLMs. Given the set of candidate relation types output $\mathcal{M}_g(\boldsymbol{s}_g)$ by prompting *relation generator*, we concatenate $\mathcal{M}_g(\boldsymbol{s}_g)$ with the task description prompt $\boldsymbol{s}_d$ and pass the combined prompt to the *relation discriminator*.

The overall process can be formulated as obtaining a relation set $\mathbf{R} = \{\mathcal{R}_1, \mathcal{R}_2, ..., \mathcal{R}_R\}$ from the two-stage LLM outputs, represented as $\mathcal{M}_d(\{\boldsymbol{s}_d\} \cup \mathcal{M}_g(\boldsymbol{s}_g))$, where $\mathcal{R}_r$ represents the textual description of $r$-th semantic relation.

## 4.2 Semantic edge decomposition

Given the set of semantic relation types $\mathbf{R}$ identified in Section 4.1, we deploy an LLM-based *relation decomposer* $\mathcal{M}_c$ tasked with assigning relevant relations to each edge $(i, j)$. A major advantage of utilizing LLMs in this context is their capability to perform multi-label classification, useful in realistic scenarios where a single edge often convey multiple semantic meanings. For instance, in an IMDB graph, two connected movie nodes might share both a common director and actor. Reflecting such real-world complexities, we instruct $\mathcal{M}_c$ to determine all possible relations that the given edge can be categorized under. Equipped with raw texts $t_i$ and $t_j$ associated with nodes $v_i$ and $v_j$, the decomposition process is expressed as $\mathcal{M}_c(\{\boldsymbol{s}_c\} \cup \{t_i, t_j\})$ with $\boldsymbol{s}_c$ indicating the instruction prompt for $\mathcal{M}_c$.

## 4.3 Integration with conventional GNNs

The edges disentangled by the *relation decomposer* can be flexibly integrated into either multi-relational GNNs (Schlichtkrull et al., 2018; Wang et al., 2019; Yang et al., 2023) or edge-featured GNNs (Hu et al., 2020c; Shi et al., 2020; Rampášek et al., 2022), highlighting its versatility.

**Multi-relational GNNs.** When paired with multi-relational GNNs, the decomposed edges categorized into $R$ types of relations are treated as $R$ distinct sub-structures $\{\mathcal{E}_1, \mathcal{E}_2, ..., \mathcal{E}_R\}$. When a single edge is assigned with multiple relation types, it is included in several corresponding $\mathcal{E}_r$. Each set $\mathcal{E}_r$ is utilized to perform type-specific neighborhood aggregation. For a given node $i$ at the $l$-th layer, these multi-relational GNNs are mathematically formulated as follows:

$$\boldsymbol{h}_i^{(l+1)} = \psi_{\text{rel}}\left(\boldsymbol{h}_i^{(l)}, \ \{\texttt{AGG}(\{\boldsymbol{h}_j^{(l)}, \forall j \in \mathcal{N}_i^{(r)}\})\}_{r=1}^R\right), \tag{3}$$

where $\mathcal{N}_i^{(r)}$ denotes the set of neighbors of $i$ connected via type-$r$ relation. Here, $\psi_{\text{rel}}$ represents the update function that combines outputs from edge type-wise aggregation (and optionally, the hidden representation of itself (Schlichtkrull et al., 2018)). In general, $\psi_{\text{rel}}$ is implemented using mean, (weighted) sum, or attention operators.

**Edge-featured GNNs.** In addition, the decomposed edges facilitated by **RoSE** can be incorporated as edge features for edge-featured GNNs. Specifically, given relation type descriptions $\mathbf{R} = \{\mathcal{R}_1, \mathcal{R}_2, ..., \mathcal{R}_R\}$ curated from *relation generator* and *discriminator*, we utilize the same PLM $g_\phi$ employed for encoding node features to embed each type description $\mathcal{R}_r$, yielding a set of relational features. Subsequently, for each edge $(i, j)$, the edge feature $\boldsymbol{e}_{ij}$ is assigned as the relational feature corresponding to the specific relation type associated with that edge, as determined by the *relation decomposer*. In cases where multiple edge types are applicable to a single edge, we

incorporate all relevant edge features by duplicating the edge with each corresponding type. The operations for an individual node $i$ at the $l$-th layer in edge-featured GNNs are formulated as follows:

$$\boldsymbol{h}_i^{(l+1)} = \psi\left(\boldsymbol{h}_i^{(l)},\ \mathtt{AGG}\big(\{\boldsymbol{h}_j^{(l)}, \xi^{(l+1)}(\boldsymbol{e}_{ij})|\forall j \in \mathcal{N}_i\}\big)\right), \tag{4}$$

where $\xi^{(l+1)}$ denotes a function that linearly maps $\boldsymbol{e}_{uv}$ to the same representational space as $\boldsymbol{h}_u^{(l)}$.

### 4.4 EFFICIENT RELATION TYPE ANNOTATION

When dealing with graphs with dense edges, the number of edges to be annotated significantly increases, which may incur expensive costs when using non-free LLMs as the backbone. To this end, we introduce an efficient node-wise query edge sampling strategy that reduces the number of queries required for LLM-based relation type classification. We assume that neighboring nodes $j_1$ and $j_2$ of a node $i$, which are close in the feature space, are likely to have similar semantic relationships with $i$. Building upon this intuition, for each node $i$, we randomly traverse its neighbors and query their relationships until either (i) all kinds of edge types are discovered or (ii) a predefined patience threshold $\gamma$ for per-node LLM queries is reached. For the remaining unqueried neighbors, we find their closest annotated neighbor and assign the same relation types as the corresponding annotation, akin to a pseudo-labeling approach. This approach can greatly reduce the number of queries associated with LLM-based edge classification, particularly on graphs with dense edges. The overall procedures is detailed in Algorithm 1. We illustrate the performance and efficiency of this approach in large-scale experiments and Appendix B.

---

**Algorithm 1** Efficient Relation Type Annotation

1: **Input:** Node $i$, Neighborhood $\mathcal{N}_i$
2: **Output:** List of relationship labels $\mathbf{L}$
3:
4: $\mathbf{S_{ng}} \leftarrow []$     # List of encountered neighbors
5: $\mathbf{S_{lb}} \leftarrow []$     # Labels of encountered edges
6: $c \leftarrow 0$             # Initialize patience
7: **for** $j$ in $\mathcal{N}_i$ **do**
8:     **if** $(|\mathrm{Set}(\mathbf{S_{lb}})| \geq R)$ **or** $(c \geq \gamma)$ **then**
9:        # Upon satisfying (i) or (ii), escape
10:        **break**
11:     **else**
12:        Add $j$ to $\mathbf{S_{ng}}$
13:        Add $\mathcal{M}_c\left(\{\boldsymbol{s}_c\} \cup \{t_i, t_j\}\right)$ to $\mathbf{S_{lb}}$
14:        $c \leftarrow c + 1$
15:     **end if**
16: **end for**
17:
18: # Initialize with labels of encountered edges
19: $\mathbf{L} \leftarrow \mathbf{S_{lb}}$
20: **for** $u$ in $\mathcal{N}_i \setminus \mathrm{Set}(\mathbf{S_{ng}})$ **do**
21:     $l \leftarrow \mathrm{argmin}_{v \in \{0,1,\dots,|\mathbf{S_n}|\}}\left(\mathrm{dist}(\mathbf{S_{ng}}[v], u)\right)$
22:     Add $\mathbf{S_{lb}}[l]$ to $\mathbf{L}$
23: **end for**

---

## 5 EXPERIMENTS

In our experiments, we evaluate our proposed framework on the node classification task using seven well-established benchmarks: Cora (McCallum et al., 2000), Pubmed (Sen et al., 2008), WikiCS (Mernyei & Cangea, 2007), IMDB (Fu et al., 2020), Cornell, Texas, and Wisconsin (Craven et al., 1998). To assess the effectiveness of our approach, we compare **RoSE** with a wide range of existing GNN architectures, including both traditional and popular GNNs (Kipf & Welling, 2016; Veličković et al., 2017; Xu et al., 2018; Schlichtkrull et al., 2018; Wang et al., 2019; Hu et al., 2020c), as well as transformer-based GNNs (Shi et al., 2020; Rampášek et al., 2022; Yang et al., 2023). The GNNs considered in our experiments can be broadly categorized as (1) Multi-relational GNNs, such as RGCN (Schlichtkrull et al., 2018), HAN (Wang et al., 2019), and SeHGNN (Yang et al., 2023); (2) Edge-featured GNNs, including GIN (Hu et al., 2020c), UniMP (Shi et al., 2020), and GraphGPS (Rampášek et al., 2022); and (3) Single-type edge processing GNNs, such as GCN (Kipf & Welling, 2016), GAT (Veličković et al., 2017), and JKNet (Xu et al., 2018). For the edge decomposition in our framework, we adopted LLaMA3-8b and 70b (Touvron et al., 2023) as foundational LLMs. Detailed dataset descriptions and experimental configurations are specified in Appendix C.

### 5.1 MAIN RESULTS

Table 2 presents the node classification accuracy results of integrating various GNN architectures with our proposed **RoSE**, across various datasets. The experiments demonstrate that our method achieves

Table 2: Node classification accuracy (%) on various datasets and GNN architectures, averaged over 10 runs (± SEM). The best and second best performances are represented by **bold** and underline.

| Type | Model | Pubmed | IMDB | Cornell | Texas | Wisconsin | Cora | WikiCS | Avg Gain |
|---|---|---|---|---|---|---|---|---|---|
| **Single-type** | **GCN** | 89.32 ± 0.11 | 64.04 ± 0.43 | 48.20 ± 2.18 | 62.94 ± 2.49 | 51.56 ± 1.79 | 88.05 ± 0.40 | 82.58 ± 0.27 | - |
| | **GAT** | 88.64 ± 0.11 | 64.39 ± 0.44 | 57.00 ± 1.56 | 66.86 ± 1.48 | 56.25 ± 2.29 | 87.74 ± 0.38 | 82.79 ± 0.16 | - |
| | **JKNet** | 89.68 ± 0.14 | 63.00 ± 0.54 | 56.00 ± 1.52 | 61.57 ± 2.92 | 57.50 ± 1.19 | 87.16 ± 0.41 | 82.94 ± 0.28 | - |
| **Multi-relational** | **RGCN** | 87.98 ± 0.14 | 62.96 ± 0.44 | 57.60 ± 1.78 | 65.88 ± 1.86 | 59.22 ± 1.70 | 88.01 ± 0.47 | 82.02 ± 0.23 | - |
| | **+ RoSE (8b)** | 90.23 ± 0.10 | 67.77 ± 0.60 | 61.40 ± 2.06 | 71.96 ± 1.82 | 70.78 ± 1.45 | 90.28 ± 0.45 | 86.81 ± 0.16 | + 5.08 |
| | **+ RoSE (70b)** | 89.68 ± 0.14 | **71.57 ± 0.42** | 63.80 ± 1.86 | 73.53 ± 1.42 | 75.31 ± 1.48 | **91.77 ± 0.38** | **88.52 ± 0.19** | **+ 7.22** |
| | **HAN** | 88.68 ± 0.15 | 63.24 ± 0.54 | 56.00 ± 1.67 | 68.82 ± 2.12 | 58.28 ± 1.99 | 87.55 ± 0.37 | 83.32 ± 0.26 | - |
| | **+ RoSE (8b)** | 90.09 ± 0.15 | 66.83 ± 0.48 | 60.00 ± 1.47 | 72.94 ± 1.64 | 72.50 ± 1.78 | 89.23 ± 0.28 | 86.12 ± 0.15 | + 4.55 |
| | **+ RoSE (70b)** | 89.77 ± 0.12 | 69.55 ± 0.43 | 62.80 ± 1.86 | 72.94 ± 1.58 | 74.38 ± 1.49 | 90.31 ± 0.38 | 87.49 ± 0.15 | + 5.91 |
| | **SeHGNN** | 87.97 ± 0.19 | 62.72 ± 0.52 | 60.00 ± 1.30 | 71.37 ± 1.28 | 65.31 ± 1.95 | 86.58 ± 0.39 | 82.53 ± 0.19 | - |
| | **+ RoSE (8b)** | 89.93 ± 0.18 | 68.27 ± 0.51 | 62.00 ± 1.41 | 73.33 ± 1.86 | 77.34 ± 1.04 | 89.53 ± 0.32 | 86.94 ± 0.18 | + 4.41 |
| | **+ RoSE (70b)** | 89.50 ± 0.23 | 70.99 ± 0.50 | 64.60 ± 2.12 | **77.45 ± 1.15** | 76.09 ± 1.31 | 91.38 ± 0.50 | 87.96 ± 0.20 | + 5.93 |
| **Edge-featured** | **UniMP** | 89.92 ± 0.16 | 69.98 ± 0.58 | 63.40 ± 1.79 | 71.18 ± 2.00 | 78.44 ± 1.50 | 87.20 ± 0.59 | 84.29 ± 0.23 | - |
| | **+ RoSE (8b)** | 90.21 ± 0.12 | 69.55 ± 0.62 | 67.80 ± 2.13 | 76.08 ± 1.79 | **80.94 ± 1.12** | 89.17 ± 0.54 | 86.33 ± 0.21 | + 2.24 |
| | **+ RoSE (70b)** | **90.37 ± 0.18** | 70.41 ± 0.64 | 67.80 ± 1.78 | 76.47 ± 1.73 | 79.84 ± 1.54 | 89.52 ± 0.41 | 87.69 ± 0.18 | + 2.52 |
| | **GIN** | 89.77 ± 0.15 | 67.59 ± 0.41 | 64.60 ± 2.08 | 68.63 ± 1.73 | 73.28 ± 2.06 | 87.05 ± 0.36 | 83.03 ± 0.21 | - |
| | **+ RoSE (8b)** | 89.68 ± 0.15 | 68.27 ± 0.69 | **68.20 ± 1.48** | 74.51 ± 2.13 | 79.22 ± 1.19 | 88.55 ± 0.30 | 83.32 ± 0.29 | + 2.54 |
| | **+ RoSE (70b)** | 89.55 ± 0.15 | 69.12 ± 0.68 | 66.20 ± 1.18 | 72.75 ± 1.45 | 77.03 ± 2.05 | 88.93 ± 0.32 | 84.84 ± 0.17 | + 2.07 |
| | **GraphGPS** | OOM | 66.85 ± 0.48 | 60.80 ± 1.73 | 70.20 ± 1.84 | 74.53 ± 0.77 | 85.14 ± 0.45 | 83.05 ± 0.26 | - |
| | **+ RoSE (8b)** | OOM | 67.69 ± 0.56 | 66.60 ± 1.88 | 73.14 ± 2.13 | 76.56 ± 1.90 | 87.53 ± 0.30 | 83.48 ± 0.23 | + 2.41 |
| | **+ RoSE (70b)** | OOM | 68.48 ± 0.54 | 64.00 ± 1.60 | 72.75 ± 2.24 | 77.34 ± 1.49 | 88.10 ± 0.45 | 85.24 ± 0.17 | + 2.56 |

marked improvements in accuracy across multi-relational GNN architectures. Notably, lightweight architectures such as RGCN and HAN, when integrated with **RoSE**, achieve performance comparable to complex transformer-based architectures like UniMP and GraphGPS. For instance, on the WikiCS dataset, RGCN with **RoSE** surpasses the vanilla UniMP architecture, setting a new state-of-the-art performance. Edge-featured architectures also exhibit significant improvements, with gains of up to 6% on Texas and Wisconsin datasets with GIN.

It is worth emphasizing that the integration of **RoSE** consistently enhances performance in 40 out of 41 settings, regardless of the original accuracy. Particularly impressive improvements are observed on datasets such as IMDB, Cornell, Texas, and Wisconsin, where GNNs have typically struggled. These results underscore the versatility of **RoSE** in improving node classification performance, irrespective of the original dataset composition. Furthermore, the scalability of **RoSE** with larger language models (e.g., **RoSE** 70b) is evident, further boosting performance in most scenarios, highlighting the effectiveness of leveraging advanced reasoning capabilities within the proposed pipeline.

Table 3: Semantic relation types generated from the *relation generator* and filtered from the *relation discriminator*. Short description of each relation is highlighted in **bold** and underline.

| Semantic Relations of Cora Dataset | |
|---|---|
| **Retained Relations** | **Filtered Relations** |
| • **Methodology Similarity**: Link papers that utilize similar methodological approaches, algorithms, or architectures to tackle their research objectives. This groups papers based on their technical commonalities. | • **Problem Similarity**: Connect papers that address similar research problems or questions, even if they use different approaches. This captures papers that are thematically related. |
| • **Contrasting Approaches**: Connect papers that explore divergent or contrasting approaches to a similar problem. This could surface insightful comparisons and foster a more holistic understanding of the problem space. | • **Performance Benchmark**: Associate papers that utilize the same benchmark dataset, evaluation metric, or performance comparison framework. This allows for standardized comparisons across models. |
| • **Theoretical Foundation**: Link papers that build upon the same fundamental theories, principles or mathematical formulations. This traces the theoretical lineage and underpinnings across papers. | • **Shared Challenges**: Group papers that grapple with similar challenges, limitations or open problems yet to be fully addressed. This synthesizes common hurdles faced by different techniques. |
| • **Sequential Refinement**: Connect papers where one incrementally improves or optimizes the techniques proposed by the other. This captures the evolutionary trajectory of methods within a research area. | • **Conceptual Parallels**: Link papers that draw conceptual parallels, analogies or inspiration from techniques in other domains and adapt them to the problem at hand. This captures cross-pollination of ideas. |
| • **Shared Application Domain**: Associate papers that apply their techniques to the same application domain or real-world problem, such as image classification, natural language processing, robotics, etc. This highlights practical use-case similarities. | • **Complementary Insights**: Connect papers that offer complementary insights, where the findings of one augment the understanding or interpretation of the results in another. This provides a more comprehensive picture. |

## 5.2 Additional experiments

Table 5: Node classification accuracy (%) and computation time analysis on large-scale datasets, averaged over 10 runs ($\pm$ SEM). The best and second best performances are represented by **bold** and underline. The computation time was measured using NVIDIA GeForce RTX 3090 GPU/Intel(R) Xeon(R) Gold 5215 CPU @2.50GHz and LLaMA3 8b Instruct with 8-bit quantization.

| Type | Model | Amazon-History | OGBN-Products | OGBN-Arxiv | Avg Gain |
|---|---|---|---|---|---|
| **Multi-relational** | **RGCN** | $81.27 \pm 0.13$ | $69.34 \pm 0.09$ | $68.31 \pm 0.03$ | - |
| | **+ RoSE-efficient (8b)** | $84.87 \pm 0.09$ | $\underline{74.25 \pm 0.19}$ | $73.35 \pm 0.05$ | + 4.52 |
| | **+ RoSE-original (8b)** | $85.06 \pm 0.11$ | $\mathbf{75.26 \pm 0.17}$ | $\underline{73.82 \pm 0.05}$ | + 5.07 |
| | **HAN** | $81.78 \pm 0.12$ | $69.29 \pm 0.11$ | $68.82 \pm 0.06$ | - |
| | **+ RoSE-efficient (8b)** | $84.98 \pm 0.12$ | $73.26 \pm 0.32$ | $73.78 \pm 0.05$ | + 4.04 |
| | **+ RoSE-original (8b)** | $85.00 \pm 0.12$ | $74.02 \pm 0.22$ | $73.80 \pm 0.06$ | + 4.31 |
| | **SeHGNN** | $81.89 \pm 0.11$ | $66.59 \pm 0.08$ | $68.90 \pm 0.06$ | - |
| | **+ RoSE-efficient (8b)** | $\underline{85.38 \pm 0.10}$ | $72.04 \pm 0.20$ | $73.41 \pm 0.06$ | + 4.48 |
| | **+ RoSE-original (8b)** | $\mathbf{85.49 \pm 0.13}$ | $73.00 \pm 0.11$ | $\mathbf{74.03 \pm 0.04}$ | + 5.05 |
| **Edge-featured** | **UniMP** | $80.32 \pm 0.11$ | $68.87 \pm 0.10$ | OOM | - |
| | **+ RoSE-efficient (8b)** | $83.78 \pm 0.42$ | $72.84 \pm 0.15$ | OOM | + 3.72 |
| | **+ RoSE-original (8b)** | $84.19 \pm 0.10$ | $73.59 \pm 0.20$ | OOM | + 4.30 |
| | **GIN** | $81.54 \pm 0.14$ | $63.09 \pm 0.07$ | $64.95 \pm 0.09$ | - |
| | **+ RoSE-efficient (8b)** | $82.90 \pm 0.14$ | $71.98 \pm 0.25$ | $70.89 \pm 0.08$ | $\underline{+ 5.40}$ |
| | **+ RoSE-original (8b)** | $83.67 \pm 0.09$ | $72.21 \pm 0.12$ | $73.20 \pm 0.06$ | $\mathbf{+ 6.50}$ |
| | **GraphGPS** | OOM | OOM | OOM | - |
| | **+ RoSE-efficient (8b)** | OOM | OOM | OOM | - |
| | **+ RoSE-original (8b)** | OOM | OOM | OOM | - |
| **#(Queries)** | **RoSE-original (8b)** | 358,574 | 74,420 | 1,166,243 | - |
| | **RoSE-efficient (8b)** | **58,545 (16.3%)** | **24,024 (32.2%)** | **480,014 (41.1%)** | + 29.87 |
| **Duration (min.)** | **RoSE-original (8b)** | 199.12 | 43.04 | 686.73 | - |
| | **RoSE-efficient (8b)** | **32.71 (16.4%)** | **13.49 (31.3%)** | **277.16 (40.4%)** | + 29.37 |

**Effect of relation discriminator.** In this experiment, we analyze the necessity and effectiveness of *relation discriminator*. We begin with a case study on the Cora dataset to demonstrate its necessity. Then, we perform an ablation study on node classification performance on Cora and Texas datasets with and without *relation discriminator* to exhibit its effectiveness.

Table 3 presents the set of retained and excluded relation types from the Cora co-citation dataset, where nodes represent scientific publications with paper abstracts as their text attribute. The relations curated from *relation generator* are generally plausible; however, some generated types are either difficult to determine through textual

Table 4: Step-wise evaluation on Texas and Cora in comparison without *relation discriminator*, averaged over 10 runs ($\pm$ SEM). The best and second-best performances are represented by **bold** and underline.

| GNNs | | LLaMA3 8b | | LLaMA3 70b | | |
|---|---|---|---|---|---|---|
| | | Texas | Cora | Texas | Cora | Avg Gain |
| **RGCN** | w/o $\mathcal{M}_d$ | $70.00 \pm 2.27$ | $87.66 \pm 0.42$ | $73.14 \pm 1.39$ | $87.94 \pm 0.42$ | |
| | **RoSE** | $71.96 \pm 1.82$ | $\mathbf{90.28 \pm 0.45}$ | $73.53 \pm 1.42$ | $\mathbf{91.77 \pm 0.38}$ | + 2.20 |
| **HAN** | w/o $\mathcal{M}_d$ | $71.37 \pm 1.47$ | $86.23 \pm 0.31$ | $71.57 \pm 1.69$ | $86.52 \pm 0.40$ | |
| | **RoSE** | $72.94 \pm 1.64$ | $89.23 \pm 0.28$ | $72.94 \pm 1.58$ | $90.31 \pm 0.38$ | + 2.43 |
| **SeHGNN** | w/o $\mathcal{M}_d$ | $72.54 \pm 1.49$ | $86.15 \pm 0.47$ | $74.51 \pm 1.92$ | $86.98 \pm 0.38$ | |
| | **RoSE** | $73.33 \pm 1.86$ | $\underline{89.53 \pm 0.32}$ | $\mathbf{77.06 \pm 0.68}$ | $91.38 \pm 0.50$ | $\underline{+ 2.78}$ |
| **UniMP** | w/o $\mathcal{M}_d$ | $73.92 \pm 2.59$ | $87.55 \pm 0.49$ | $75.10 \pm 1.67$ | $87.40 \pm 0.50$ | |
| | **RoSE** | $\mathbf{76.08 \pm 1.79}$ | $89.17 \pm 0.54$ | $\underline{76.47 \pm 1.73}$ | $89.52 \pm 0.41$ | + 1.82 |
| **GIN** | w/o $\mathcal{M}_d$ | $70.59 \pm 2.20$ | $86.85 \pm 0.41$ | $69.61 \pm 1.58$ | $86.52 \pm 0.41$ | |
| | **RoSE** | $\underline{74.51 \pm 2.13}$ | $88.55 \pm 0.30$ | $72.75 \pm 1.45$ | $88.93 \pm 0.32$ | $\mathbf{+ 2.79}$ |
| **GraphGPS** | w/o $\mathcal{M}_d$ | $73.33 \pm 1.65$ | $85.76 \pm 0.19$ | $70.39 \pm 2.90$ | $86.72 \pm 0.50$ | |
| | **RoSE** | $73.14 \pm 2.13$ | $87.53 \pm 0.30$ | $72.75 \pm 2.24$ | $88.10 \pm 0.45$ | + 1.33 |

analysis of node attributes or exhibit significant overlap with each other. For instance, the relation type Performance Benchmark (second relation in the rightmost column) is not easily identified based on paper abstracts, as these abstracts often do not enumerate each benchmark used within the paper. Thus, determining such relations exceeds the capability of language models. Additionally, Complementary Insights (last element of the filtered relations) overlaps significantly with Contrasting Approaches, introducing redundancy. Consequently, such relations are filtered out by the *relation discriminator*. Further case studies and of relation types and decomposed examples are provided in Appendix B.

We also empirically validate the efficacy of this filtration on the Texas and Cora datasets by evaluating the node classification performance with and without the *relation discriminator*, as shown in Table 4. Consistent improvements are observed with *relation discriminator* across 23 out of 24 settings, showing an average 2.23% increase in accuracy.

**Effect of relation decomposer.** Table 6 compares the performance of **RoSE** with rule-based decomposition methods on the IMDB, Texas, and Cora datasets. The baselines are formulated as follows: (1) **Random**, which randomly decomposes edges into different relations; (2) **Distance**,

Table 6: Node classification accuracy (%) on IMDB, Texas, and Cora with multi-relational and edge-featured GNNs, averaged over 10 runs (± SEM). The best and second best performances for each architecture are represented by **bold** and underline.

| Multi-relational GNNs | | IMDB | Texas | Cora | Edge-featured GNNs | | IMDB | Texas | Cora |
|---|---|---|---|---|---|---|---|---|---|
| | Random | $62.90 \pm 0.50$ | $66.47 \pm 1.67$ | $87.00 \pm 0.29$ | | Random | $68.65 \pm 0.40$ | $71.18 \pm 1.90$ | $87.02 \pm 0.30$ |
| | Distance | $66.99 \pm 0.48$ | $66.67 \pm 2.15$ | $88.03 \pm 0.46$ | | Distance | $69.12 \pm 0.68$ | $72.94 \pm 1.88$ | $87.94 \pm 0.41$ |
| RGCN | **RoSE** (8b) | $67.77 \pm 0.60$ | $71.96 \pm 1.82$ | $\underline{90.28} \pm 0.45$ | UniMP | **RoSE** (8b) | $69.55 \pm 0.62$ | $\underline{76.08} \pm 1.79$ | $89.17 \pm 0.54$ |
| | **RoSE** (70b) | **71.57** $\pm$ **0.42** | $\underline{73.53} \pm 1.42$ | **91.77** $\pm$ **0.38** | | **RoSE** (70b) | **70.41** $\pm$ **0.64** | **76.47** $\pm$ **1.73** | **89.52** $\pm$ **0.41** |
| | G.T. | $\underline{68.66} \pm 0.57$ | **76.47** $\pm$ **1.82** | - | | G.T. | $\underline{69.87} \pm 0.57$ | $77.84 \pm 1.94$ | - |
| | Random | $62.76 \pm 0.59$ | $67.65 \pm 1.85$ | $86.19 \pm 0.42$ | | Random | $67.23 \pm 0.42$ | $69.22 \pm 1.90$ | $79.96 \pm 0.93$ |
| | Distance | $66.66 \pm 0.50$ | $68.63 \pm 2.09$ | $87.13 \pm 0.49$ | | Distance | $68.27 \pm 0.37$ | $70.59 \pm 1.96$ | $86.92 \pm 0.50$ |
| HAN | **RoSE** (8b) | $66.83 \pm 0.48$ | **72.94** $\pm$ **1.64** | $\underline{89.23} \pm 0.28$ | GIN | **RoSE** (8b) | $68.27 \pm 0.69$ | **74.51** $\pm$ **2.13** | $\underline{88.55} \pm 0.30$ |
| | **RoSE** (70b) | **69.55** $\pm$ **0.43** | **72.94** $\pm$ **1.58** | **90.31** $\pm$ **0.38** | | **RoSE** (70b) | **69.12** $\pm$ **0.68** | $72.75 \pm 1.45$ | **88.93** $\pm$ **0.32** |
| | G.T. | $\underline{68.39} \pm 0.62$ | $71.37 \pm 2.24$ | - | | G.T. | $\underline{68.54} \pm 0.43$ | $\underline{74.12} \pm 1.59$ | - |
| | Random | $62.46 \pm 0.56$ | $70.98 \pm 2.09$ | $86.00 \pm 0.36$ | | Random | $67.23 \pm 0.44$ | $69.41 \pm 2.15$ | $85.80 \pm 0.25$ |
| | Distance | $67.97 \pm 0.43$ | $71.57 \pm 1.15$ | $87.07 \pm 0.32$ | | Distance | $66.98 \pm 0.75$ | $69.22 \pm 1.76$ | $86.46 \pm 0.44$ |
| SeHGNN | **RoSE** (8b) | $68.27 \pm 0.51$ | $73.33 \pm 1.86$ | $\underline{89.53} \pm 0.32$ | GraphGPS | **RoSE** (8b) | $\underline{67.69} \pm 0.56$ | **73.14** $\pm$ **2.13** | $\underline{87.53} \pm 0.30$ |
| | **RoSE** (70b) | **70.99** $\pm$ **0.44** | $\underline{77.45} \pm 1.15$ | **91.38** $\pm$ **0.50** | | **RoSE** (70b) | **68.48** $\pm$ **0.54** | $\underline{72.75} \pm 2.24$ | **88.10** $\pm$ **0.45** |
| | G.T. | $\underline{69.00} \pm 0.48$ | **78.04** $\pm$ **1.07** | - | | G.T. | $67.07 \pm 0.78$ | $\underline{72.75} \pm 1.70$ | - |

which decomposes edges into two relations based on the cosine distance between the associated node features obtained from pre-trained language models (PLMs), categorizing them as semantically similar or different edges. The ground-truth decomposition (**GT**) obtained through manual annotation is also presented for comparison. It is important to note that the ground-truth decomposition consists of mutually exclusive relations, and for the Cora dataset, ground truth information is not available.

The results demonstrate the superior performance of **RoSE** compared to basic rule-based methods, highlighting the necessity of leveraging LLMs for intricate semantic decomposition. Moreover, **RoSE** achieves the best or second-best performance on all ablative datasets, even when compared to the ground truth decomposition. This underscores the effectiveness of our *relation decomposer* component, which identifies all relations that accurately describe a given edge, thereby providing a richer source of information for GNN architectures to exploit.

**Scalability to large-scale datasets.** We extended our evaluations on large-scale datasets for **RoSE** (**RoSE**-original) and **RoSE** with the efficient query technique (**RoSE**-efficient) in Table 5. The benchmark datasets employed in this study include Amazon-History (Yan et al., 2023), a subset of OGBN-Products, and OGBN-arXiv (Hu et al., 2020b). Across these datasets, both **RoSE**-original and **RoSE**-efficient demonstrated consistent performance improvements, achieving average enhancements of 4.48% and 5.10%, respectively.

Furthermore, we compared the total number of queries sent to the relation decomposer by **RoSE**-original and **RoSE**-efficient. The results indicate that **RoSE**-efficient reduces the number of queries by up to 41%, underscoring its efficiency while maintaining robust performance. Notably, these improvements are realized through single-round LLM queries, thereby eliminating the need for re-computation or additional fine-tuning required by previous LLM-based feature enhancement methods (He et al., 2023; Duan et al., 2023; Chien et al., 2021). Consequently, the scalability of **RoSE** allows practitioners to select an LLM that aligns with their computational constraints without compromising the method's effectiveness.

**Comparison with node feature enhancement methods.** To further validate the effectiveness of our proposed method, we compared **RoSE** against recent node feature enhancement techniques, specifically TAPE (He et al., 2023) and OFA (Liu et al., 2023a), utilizing RGCN as the backbone. The experiments were conducted on the same datasets employed by the baseline

Table 7: Node classification accuracy (%) of **RoSE**, TAPE, and OFA, averaged over 10 runs. The best performance in each architecture is represented by **bold**.

| Setting | Methods | Cora | Pubmed | WikiCS |
|---|---|---|---|---|
| **w/ Deberta node feature** | TAPE | $90.15 \pm 0.32$ | $89.42 \pm 0.17$ | $82.81 \pm 0.24$ |
| | **RoSE** (8b) | **92.47** $\pm$ **0.50** | **95.59** $\pm$ **0.16** | **91.78** $\pm$ **0.36** |
| **w/ sparse split** | OFA | $75.90 \pm 1.26$ | $75.54 \pm 0.05$ | $78.34 \pm 0.35$ |
| | **RoSE** (8b) | **84.37** $\pm$ **0.90** | **79.05** $\pm$ **0.71** | **82.35** $\pm$ **0.28** |

methods, namely Cora, Pubmed, and WikiCS. Since TAPE employs a fine-tuned DeBERTa as the feature encoder for nodes' text attributes, we also adopted the same model to encode the original node attributes. Additionally, we evaluate **RoSE** under a sparse label setting when comparing with OFA, adhering to the experimental configuration outlined in Liu et al. (2023a).

As presented in Table 7, our method outperforms TAPE across all datasets, achieving a maximum improvement of 8.97% on WikiCS. Furthermore, **RoSE** consistently exceeds the performance of OFA, with an average improvement of 5.33%, highlighting the robustness of our method even in sparse label scenarios. These results demonstrate the efficacy of decomposing edges into multiple semantic relations, outperforming methods that rely on LLM-enhanced node features. Additionally, we provide a comparison with the graph prompting approach in Appendix B.

## 6 RELATED WORKS

**Node feature-level enhancement.**   The presence of textual content in TAGs has inspired researchers to explore beyond traditional feature encoding methods such as bag-of-words (Harris, 1954) and skip-grams Mikolov et al. (2013). Consequently, numerous studies have been proposed to generate semantically rich node features by employing relatively smaller pretrained language models (PLMs) (Yang et al., 2021; Chien et al., 2021; Zhao et al., 2022; Dinh et al., 2023), including DeBERTa (He et al., 2020), Sentence-BERT (Reimers & Gurevych, 2019), E5 (Wang et al., 2022), and OpenAI's text-ada-embedding-002 (Neelakantan et al., 2022), alongside larger LLMs such as GPT (Brown et al., 2020) and LLaMA (Touvron et al., 2023). These efforts can be broadly categorized into three approaches: **(1) Cascading structure** receives initial node features from the output embeddings of PLMs and LLMs, followed by the deployment of GNNs to obtain final representations. This independent framework has been widely adopted across various studies in TAG literature (Zhou et al., 2019; Zhu et al., 2021; Li et al., 2021; Hu et al., 2020d; Liu et al., 2019; Chien et al., 2021; Duan et al., 2023; Liu et al., 2024). **(2) Co-training structure** involves the joint training of PLMs and GNNs within an interactive workflow. This facilitates a dynamic and correlated workflow of semantic information across connected nodes (Yang et al., 2021; Zhao et al., 2022; Dinh et al., 2023). **(3) Enhanced text augmentation** focuses on enriching the raw textual contents with PLMs and LLMs, such as by replacing text attributes with textual explanations generated by LLMs during its node classification (He et al., 2023) or augmenting external knowledge within a knowledge graph (Sun et al., 2019; Liu et al., 2020). However, these studies often overlook the diverse semantics inherent in graph structures and characterize edges as a binary adjacency matrix of uniform relation, thus leading to structural oversimplification. Although there exist few works aiming to enhance edge attributes (Jin et al., 2023b;a), these works are only applicable in settings where edge-attributed texts and ground truth relation types exist.

**LLMs with graph structural information.**   Another line of research investigates the potential of LLMs for addressing graph problems by injecting graph structural information into the input prompt of LLMs. This incorporation is achieved through various methods, including describing node adjacency in natural language (Ye et al., 2023; Guo et al., 2023; Wang et al., 2024; Fatemi et al., 2024), utilizing syntax tree into natural language representations (Zhao et al., 2023), and leveraging structural tokens (Tang et al., 2023). Although these approaches integrate structural data into LLMs, they treat graph edges as binary connections, presenting a clear distinction from our work of utilizing LLMs to automatically decompose graph structures into multiple semantic relation types.

## 7 CONCLUSION

Given the limitation of existing TAG literature in simplifying the entangled semantics in graph structure, we introduced **RoSE**, an innovative framework that leverages the analytical capabilities of LLMs to disentangle edges in a fully automated manner, based on the textual contents of connected nodes. As a pioneering effort in revealing and addressing the structural oversimplification, we believe our contributions provide valuable insights into this field. However, one limitation of our framework is its reliance on the general knowledge of LLMs for identifying relation types, which may not fully capture domain-specific relationships when applied to graphs from highly specialized domains that are not well-represented in the LLMs' training data. As future work, we plan to explore techniques such as retrieval-augmented generation (RAG) to effectively incorporate domain knowledge.

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

# Supplementary Materials

## A   Detailed prompt templates

In this section, we provide the fixed prompt templates used in our experiments for the *relation generator*, *discriminator*, and *decomposer*.

To decompose edges into various relation types, we first identify candidate semantic relation types in the given graph using a relation generator and a relation discriminator. To begin with, the relation generator produces a set of plausible candidate relations based on the following prompt components: (1) Description of what each node and edge represents, (2) A sample text attribute for a node, (3) Predefined categories of nodes, (4) Initial guidelines for generating relations. The prompt template for the *generator* used in the Cora dataset is as follows:

> **# Graph Composition Description**
> You are tasked with analyzing a graph consisting of *nodes representing papers and edges representing co-citation*.
> The predefined categories of nodes are: *[Rule Learning, Neural Networks, Case Based, Genetic Algorithms, Theory, Reinforcement Learning, Probabilistic Methods]*.
> Each paper node contains *paper abstract as a text attribute*. An example text attribute is:
>
> *Stochastic pro-positionalization of non-determinate background knowledge. : It is a well-known fact that (...)*
>
> **# Task Description**
> Your objective is to design a set of unique semantic edge types that capture meaningful relationships between the nodes based on their text attributes.
> Focus on revealing semantic connections that captures unique patterns between specific nodes. These edge types should be inferred from the textual content.
>
> Create edge types as many as you feel are absolutely necessary to decompose, while maintaining a manageable number of edge types for practical decomposition.

Subsequently, the relation discriminator filters the candidate relations generated, ensuring that only relevant and feasible relations are retained. This step addresses the noisy candidates in the initial set, which may be irrelevant or infeasible to infer given the textual attributes and the analytical capabilities of LLMs. The prompt for the relation discriminator is composed of: (1) a description of what each node and edge represents, (2) a sample text attribute for a node, (3) predefined categories of nodes, (4) preliminary guidelines for filtering candidate relations, and (5) candidate relations produced by the relation generator. The prompt template for the *discriminator* utilized in the Cora dataset is detailed below:

> **# Task Description**
> You are tasked with verifying the quality and relevance of proposed semantic edge types in a graph representing *nodes as papers and edges as co-citation*.
>
> Your objective is to identify and retain only the essential edge types for improving the performance of Graph Neural Networks (GNNs) in node classification tasks.
> The predefined categories of nodes are: *[Rule Learning, Neural Networks, Case Based, Genetic Algorithms, Theory, Reinforcement Learning, Probabilistic Methods]*.
> Each paper node contains *paper abstract as a text attribute*. An example text attribute is:
>
> *Stochastic pro-positionalization of non-determinate background knowledge. : It is a well-known fact that (...)*
>
> **# Task Requirements**
> When discriminating the edge types, consider the following guidelines:
> 1. Exclude criteria if they are beyond LLM's analytical capability such as hyperlinks, co-authorship, and common references.
> 2. Retain only criteria that are absolutely necessary for generating significant performance-enhancing edges on node classification task.
>
> **# Proposed Semantic Edge Types**
> [Relation types curated from the relation generator]

During the semantic edge decomposition phase, we query the *relation decomposer* to determine all possible relations that the given edge can be categorized under. To accomplish this, we concatenate the instruction prompt with the text attributes of the associated nodes in the input prompt for the *relation decomposer*. The input prompt template used in the Cora dataset is provided as follows:

---

**# Task Description**
You are an helpful assistant, that classifies an edge connection between two nodes into one or more of the following relation types. Note that it is a multiple-choice classification.

**# Relation Specification**
Relation types are as follows: [List of relation types]

**Node 1**: [Raw text attribute of Node 1], **Node 2**: [Raw text attribute of Node 2]
**Question**: The two nodes are connected via *co-citation*. Carefully choose relation types that likely represent the semantic relation between the two nodes.

---

Table 8: Case study of edge decomposition on the Cora dataset, classified by *relation decomposer*.

| **Classified relation types of an edge** $(v_1, v_2)$ |
|---|
| • **Methodology Similarity**: Link papers that utilize similar methodological approaches, algorithms, or architectures. 
 • **Shared Application Domain**: Associate papers that apply their techniques to the same application domain. |

| **Text attribute of node** $v_1$ |
|---|
| Stochastic pro-positionalization of non-determinate background knowledge. : It is a well-known fact that **propositional learning algorithms** require "good" features to perform well in practice. So a major step in data engineering for inductive learning is the **construction of good features** by domain experts. These features often represent properties of structured objects, where a property typically is the occurrence of a certain substructure having certain properties. To partly automate the process of "feature engineering", we devised an algorithm that searches for features which are defined by such substructures. The algorithm stochastically conducts a top-down search for first-order clauses, where each clause represents a binary feature. It differs from existing algorithms in that its search is not class-blind, and that it is capable of considering clauses ("context") of almost arbitrary length (size). Preliminary experiments are favorable, and support the view that this approach is promising. |

| **Text attribute of node** $v_2$ |
|---|
| Learning Trees and Rules with Set-valued Features. : In most learning systems examples are represented as fixed-length "feature vectors", the components of which are either real numbers or nominal values. We propose an extension of the feature-vector representation that allows the value of a feature to be a set of strings; for instance, to represent a small white and black dog with the nominal features size and species and the set-valued feature color, one might use a feature vector with size=small, species=canis-familiaris and color={white,black}. Since we make no assumptions about the number of possible set elements, this extension of the traditional feature-vector representation is closely connected to Blum's "infinite attribute" representation. We argue that many decision tree and rule learning algorithms can be easily extended to set-valued features. We also show by example that many real-world learning problems can be **efficiently and naturally represent**ed with set-valued features; in particular, text categorization problems and problems that arise in **propositionalizing** first-order representations lend themselves to set-valued features. |

## B  FURTHER ANALYSIS AND EXPERIMENTS

### B.1  CASE STUDY ON EDGE DECOMPOSITION

To verify the effectiveness of our edge decomposition, we provide examples of decomposition results on the Cora and Texas datasets. As shown in the case study on the Cora dataset, both nodes propose extensions and

Table 9: Case study of edge decomposition on the Texas dataset, classified by *relation decomposer*.

| Classified relation types of an edge $(v_3, v_4)$ |
|---|
| **Advised_By/Advises Edge**: Connects a student node and a faculty node (faculty advises or mentors that student). |
| **Text attribute of node $v_3$** |
| Simon S. Lam
Professor of Computer Sciences
Department of Computer Sciences
University of Texas Austin, Texas 78712-1188

email: lam@cs.utexas.edu
phone: (512) 471-9531
fax: (512) 471-8885
office: Taylor Hall 3.112
campus mail: Computer Science C0500

Photo and Profile
Networking Research Laboratory
CS 395T (Fall 1996)
CS 356 (Spring 1996)

Administrative Assistant (also editorial assistant for IEEE/ACM Transactions on Networking)
Kata Carbone
email: kata@cs.utexas.edu
phone: (512) 471-9524
fax: (512) 471-8885 |
| **Text attribute of node $v_4$** |
| Chung Kei Wong
last modified: Dec 11, 1996

About Me
I am a graduate student in the Department of Computer Sciences, The University of Texas at Austin. I am a member of the Networking Research Lab which is headed by Prof. Simon S. Lam.

Research Related links...
    Java Security Project
    NIST Computer Security Division
    Computer Security Resource Clearinghouse
    Role Based Access Control (RBAC)
    Prof. Ron Rivest's Cryptography and Security page

To Contact Me
EMAIL [ckwong@cs.utexas.edu]
POSTAL Computer Sciences C0500 TAY 2.124, U.T. Austin Austin TX 78712 USA |

improvements to feature representations in learning systems. Additionally, both nodes apply their techniques to the feature engineering domain. Consequently, the relation decomposer's predicted relations as "methodological similarity" and "shared application" are appropriate.

For the Texas dataset, we observe that graduate student $v_4$ is under the guidance of professor $v_3$ according to the textual contents. Therefore, the relation decomposer's predicted relation of $(v_3, v_4)$ as "Advised-By/Advises" is correct, highlighting the textual reasoning capability of the decomposer.

Table 10: Semantic relation types generated from the *relation generator* and filtered from the *relation discriminator*. Short description of each relation is highlighted in **bold** and underline.

| Semantic Relations of Texas Dataset | |
|---|---|
| **Retained Relations** | **Filtered Relations** |
| • **Teaches/Teaches_Under Edge**: Connects a faculty node and a course node (faculty teaches that course).
• **Researches/Research_Contributes_To Edge**: Connects a faculty or student node with a project node (they conduct research related to that project).
• **Advised_By/Advises Edge**: Connects a student node and a faculty node (faculty advises or mentors that student).
• **Enrolled_In/Enrolls Edge**: Connects a student node and a course node (student is enrolled in that course).
• **TA_For/Has_TA Edge**: Connects a student node and a course node (student is a teaching assistant for that course). | • **Studies_Under/Has_Student Edge**: Connects a student node to a faculty node suggesting that the student studies under that professor's guidance, without an explicit advising relationship stated.
• **Staff_Supports/Supported_By_Staff Edge**: Connects a staff node to other nodes (faculty/student/course/project) implying that the staff provides some type of administrative or technical support for that entity.
• **Affiliated_With Edge**: Connects faculty/student/staff nodes to their primary associated entity like a lab, center, department or institute mentioned in their text. |

Table 11: Node classification accuracy and the number of queries sent to *relation-decomposer* of **RoSE** and **RoSE** with efficient querying technique, averaged over 10 runs ($\pm$ SEM). The best performance in each architecture is represented by **bold**.

| GNN Architectures | | IMDB | WikiCS |
|---|---|---|---|
| **RGCN** | Vanilla | $62.96 \pm 0.44$ | $82.02 \pm 0.23$ |
| | **RoSE**-efficient (8b) | $67.22 \pm 0.33$ | $86.42 \pm 0.18$ |
| | **RoSE**-original (8b) | $\mathbf{67.77 \pm 0.60}$ | $\mathbf{86.81 \pm 0.16}$ |
| **HAN** | Vanilla | $63.24 \pm 0.54$ | $83.32 \pm 0.26$ |
| | **RoSE**-efficient (8b) | $66.52 \pm 0.64$ | $85.81 \pm 0.21$ |
| | **RoSE**-original (8b) | $\mathbf{66.83 \pm 0.48}$ | $\mathbf{86.12 \pm 0.15}$ |
| **SeHGNN** | Vanilla | $62.72 \pm 0.52$ | $82.53 \pm 0.19$ |
| | **RoSE**-efficient (8b) | $66.31 \pm 0.37$ | $86.16 \pm 0.20$ |
| | **RoSE**-original (8b) | $\mathbf{68.27 \pm 0.51}$ | $\mathbf{86.94 \pm 0.18}$ |
| **UniMP** | Vanilla | $\mathbf{69.98 \pm 0.58}$ | $84.29 \pm 0.23$ |
| | **RoSE**-efficient (8b) | $69.36 \pm 0.52$ | $86.09 \pm 0.19$ |
| | **RoSE**-original (8b) | $69.55 \pm 0.62$ | $\mathbf{86.33 \pm 0.21}$ |
| **GIN** | Vanilla | $67.59 \pm 0.41$ | $83.03 \pm 0.21$ |
| | **RoSE**-efficient (8b) | $67.15 \pm 0.56$ | $\mathbf{84.20 \pm 0.28}$ |
| | **RoSE**-original (8b) | $\mathbf{68.27 \pm 0.69}$ | $83.32 \pm 0.29$ |
| **GraphGPS** | Vanilla | $66.85 \pm 0.48$ | $83.05 \pm 0.26$ |
| | **RoSE**-efficient (8b) | $67.41 \pm 0.73$ | $\mathbf{85.14 \pm 0.18}$ |
| | **RoSE**-original (8b) | $\mathbf{67.69 \pm 0.56}$ | $83.48 \pm 0.23$ |
| **#(Queries)** | **RoSE**-original (8b) | 45698 | 215603 |
| | **RoSE**-efficient (8b) | **15391** | **40055** |
| | **Decrement** | **61.58%↓** | **78.80%↓** |

## B.2 ADDITIONAL CASE STUDIES

In extension from Section 5, we present the retained and filtered relation types for Texas datasets in Table 10. In the Texas dataset, the Studies_Under/Has_Student Edge is identified as nearly redundant with the Advised_By/Advises Edge, leading to its exclusion to avoid redundancy. Additionally, the Affiliated_With Edge is deemed too ambiguous, as it can encompass various edges generated from the Texas dataset, and is therefore removed. Hence, these findings demonstrate the effectiveness of the *relation discriminator* in identifying and filtering out relations that lack feasibility or distinctiveness, ensuring the retention of meaningful and non-redundant edges.

## B.3 EXPERIMENTS ON EFFICIENT RELATION TYPE ANNOTATION

To demonstrate the efficacy of the proposed efficient query edge sampling strategy discussed in Section 4.4, we conduct further experiments with **RoSE** using our efficient relation type annotation (denoted as **RoSE**-efficient) on graphs with the largest number of edges: WikiCS (Mernyei & Cangea, 2007) and IMDB (Fu et al., 2020). Table 11 displays the node classification performance of multi-relational and edge-featured GNNs, utilizing LLaMa3-8b (Touvron et al., 2023) as a base LLM. As demonstrated in Table 11, **RoSE**-efficient can still improve the performance of original GNNs across 10 out of 12 settings, with less than half the number of queries than **RoSE**-original. Notably, it even surpasses the performance of **RoSE** with full edge annotation (**RoSE**-original) when incorporated with GIN (Hu et al., 2020c) and GraphGPS (Rampášek et al., 2022).

Table 12: Link prediction performance (%) of **RoSE** on Cora, Pubmed, and WikiCS, averaged over 10 runs ($\pm$ SEM). The best performance in each architecture is represented by **bold**.

| Methods | Cora | | Pubmed | | WikiCS | |
|---|---|---|---|---|---|---|
| RGCN | $86.52 \pm 0.31$ | $89.00 \pm 0.25$ | $58.56 \pm 2.63$ | $75.30 \pm 1.50$ | $45.26 \pm 1.13$ | $56.02 \pm 0.15$ |
| + **RoSE** (8b) | $\mathbf{87.75} \pm \mathbf{0.67}$ | $\mathbf{92.87} \pm \mathbf{0.06}$ | $\mathbf{75.71} \pm \mathbf{0.82}$ | $\mathbf{87.20} \pm \mathbf{0.40}$ | $\mathbf{52.12} \pm \mathbf{0.55}$ | $\mathbf{66.21} \pm \mathbf{0.32}$ |

To verify the efficiency of our sampling strategy, we also compare the total number of queries sent to the *relation decomposer* by **RoSE** and **RoSE**-efficient. Remarkably, our method reduces the number of queries by more than half, while maintaining comparable performance.

## B.4 RESULTS ON LINK PREDICTION

We extended our evaluations to link prediction to further verify the versatility of **RoSE**. Using RGCN as a backbone architecture, we conducted link prediction experiments on the Cora, Pubmed, and WikiCS datasets by adopting the training/validation/test split ratio as [70, 10, 20], following the convention in Chamberlain et al. (2022). We maintained the same edge decomposition framework via LLMs and only changed the prediction head tailored for the link prediction. Specifically, we obtained the edge representations by concatenating the GNN representations of corresponding node pairs. These edge representations were then fed into a link predictor (a 2-layer MLP with a sigmoid function at the end) to predict edge existence. We evaluated the prediction performance using the Hits@K metric, which measures the proportion of ground-truth links ranked among the top K predictions. The K was set to 50 and 100.

As shown in Table Table 12, our method achieves performance improvements across all settings. Notably, we observe significant gains on the Pubmed and WikiCS datasets, with improvements of up to 17.15% and 10.19%, respectively. These results highlight the versatility of our method in link prediction tasks, as **RoSE** helps GNNs achieve better feature disentanglement.

## B.5 COMPARISON WITH GRAPH PROMPTING APPROACH

In this section, we present additional comparison of our method against graph prompting learning framework, ProG (Sun et al., 2023). We reproduced ProG under our experimental configuration of supervised learning under the same RGCN backbone and benchmarks utilized in Table 7. The results in Table 13 indicate that **RoSE** achieves superior performance compared to the graph prompting approach. We hypothesize that this notable performance gap is due to

Table 13: Node classification performance (%) of **RoSE** and ProG on Cora, Pubmed, and WikiCS, averaged over 10 runs ($\pm$ SEM). The best performance in each architecture is represented by **bold**.

| Methods | Cora | Pubmed | WikiCS |
|---|---|---|---|
| ProG | $74.24 \pm 0.77$ | $77.47 \pm 0.27$ | $64.02 \pm 0.54$ |
| **RoSE** (8b) | $\mathbf{90.28} \pm \mathbf{0.45}$ | $\mathbf{90.23} \pm \mathbf{0.10}$ | $\mathbf{70.78} \pm \mathbf{1.45}$ |

ProG's conversion of node classification tasks into graph classification tasks for multi-task learning. In the node classification task, the final node representation is obtained by performing a global pooling operation over hidden representations of all nodes in the subgraph. This may make the node representation ambiguous, as its K-hop neighbors can have diverging node labels, leading to suboptimal performance.

## B.6 SENSITIVITY TO LLM TEMPERATURE

Figure 2 compares the performance of **RoSE** with respect to the decoding temperature. Higher temperature results in higher randomness in the outputs of LLMs, and may influence the performance of the *relation decomposer*. We choose two representative GNN architectures for our evaluation, RGCN from multi-relational GNNs and GIN from edge-featured GNNs. Our experiments on IMDB, Texas, and Cora reveal that the improvements of **RoSE** are consistent across varying temperatures.

## B.7 IMPORTANCE OF SEMANTIC EDGE DECOMPOSITION - REPRESENTATIONAL ANALYSIS

We further analyze the enhancements provided by edge-decomposition strategy(presented in Section 3), in a representation learning perspective. Specifically, we analyze the UMAP visualizations of node representations obtained from RGCN (Schlichtkrull et al., 2018) and HAN (Wang et al., 2019), trained with single and multiple types of relations. Figures 3 and 4 illustrate these visualizations, each rows representing: (1) initial node features, (2) node representations learned from RGCN, and (3) node representations learned from HAN, respectively. The results demonstrate that decomposing conventional edges into multiple relation types yields more distinct, clustered representations. Conversely, simplifying the inherent and diverse semantics leads to less distinguishable representations, particularly on the WebKB datasets (Cornell, Texas, and Wisconsin) (Craven et al., 1998) when using RGCN as the backbone.

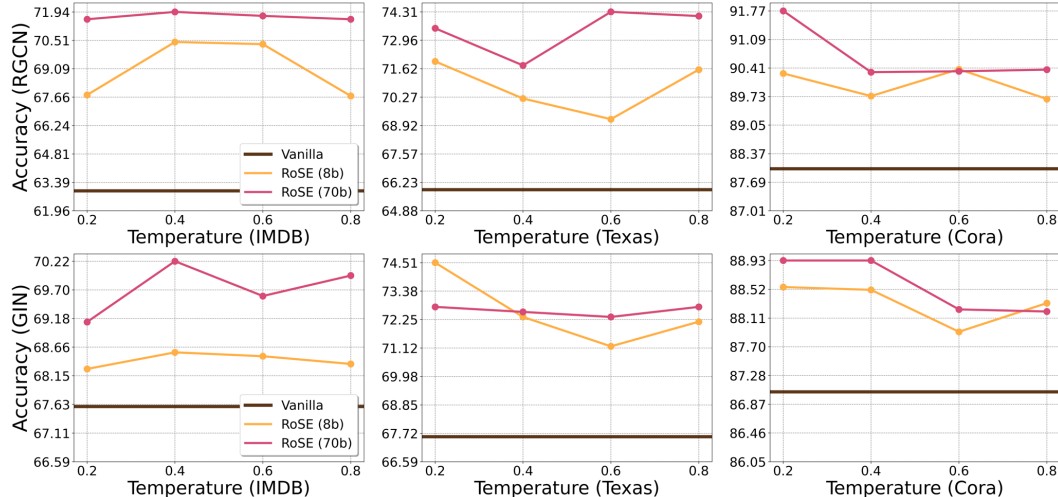

Figure 2: Sensitivity to temperature when prompting *relation decomposer*. Varied temperature (0.2 - 0.8) is denoted on the x-axis, while node classification accuracy(%) is denoted on the y-axis. Red, yellow and brown each denote **RoSE** (LLaMA3-70b), **RoSE** (LLaMA3-8b), and vanilla GNNs (RGCN and GIN), respectively.

We observe similar trends with respect to the inter-prototype similarity between representation prototypes. Specifically, we calculate per-class prototype vector $\mathbf{p}_k = \frac{1}{|C_k|} \sum_{i \in C_k} \mathbf{z}_i$, where $C_k$ denotes the set of nodes belonging to class $k$. Then we evaluate the average cosine similarity between class prototypes as $\texttt{Sim}_{\mathrm{mean}} = \mathbb{E}_{k_1 \neq k_2, \{k_1, k_2\} \subseteq C} \left( \frac{\mathbf{p}_{k_1} \cdot \mathbf{p}_{k_2}}{\|\mathbf{p}_{k_1}\| \|\mathbf{p}_{k_2}\|} \right)$, with $C$ denoting the set of class labels. Intuitively, a smaller $\texttt{Sim}_{\mathrm{mean}}$ implies more distinct class prototypes within the feature space. We plot the $\texttt{Sim}_{\mathrm{mean}}$ along the y-axis of Figure 5. As evident in the figure, our results indicate that simplifying diverse edge semantics results in less distinguishable class representations (i.e. high similarity between class prototypes). This is particularly pronounced in RGCN on Cornell and Texas dataset, where $\texttt{Sim}_{\mathrm{mean}}$ of learned representations on a single relation type is higher than inter-prototype similarities of raw features. In contrast, disentangling these semantics into multiple edge types can achieve significant improvements in inter-class separation. Specifically, for the Cornell dataset, $\texttt{Sim}_{\mathrm{mean}}$ of multi-relation type processing achieves a reduction in similarity of at least 43% across all GNNs, compared to those obtained from raw features and uniform edge type processing.

## C  EXPERIMENTAL SETTINGS

### C.1  DATASET STATISTICS

In this section, we provide an overview of the graph compositional information for our benchmark datasets:

**Pubmed** (Sen et al., 2008)  is a co-citation network in which nodes represent scientific publications and edges denote co-citations. The textual content of each node comprises the paper's abstract. The predefined categories are Diabetes Experimental, Diabetes Type I, and Diabetes Type II.

**IMDB** (Fu et al., 2020)  is a movie graph where nodes represent movies and edges indicate the overlap of movie professionals. The textual content of each node corresponds to the summarized movie description. The predefined genres are Action, Comedy, and Drama.

**WebKB**[1] (**Cornell, Texas, Wisconsin**) (Craven et al., 1998)  are hyperlink networks in which nodes represent web pages and edges are hyperlinks. The text attribute of each node represents the web page content. The predefined categories are Student, Faculty, Staff, Course, and Project.

**Cora** (McCallum et al., 2000)  is a co-citation network where nodes represent scientific papers and edges indicate co-citations. The textual content of each node comprises the paper's abstract. The predefined categories are Case-based, Genetic algorithms, Neural networks, Probabilistic methods, Reinforcement learning, Rule learning, and Theory.

---

[1] https://www.cs.cmu.edu/afs/cs.cmu.edu/project/theo-11/www/wwkb/

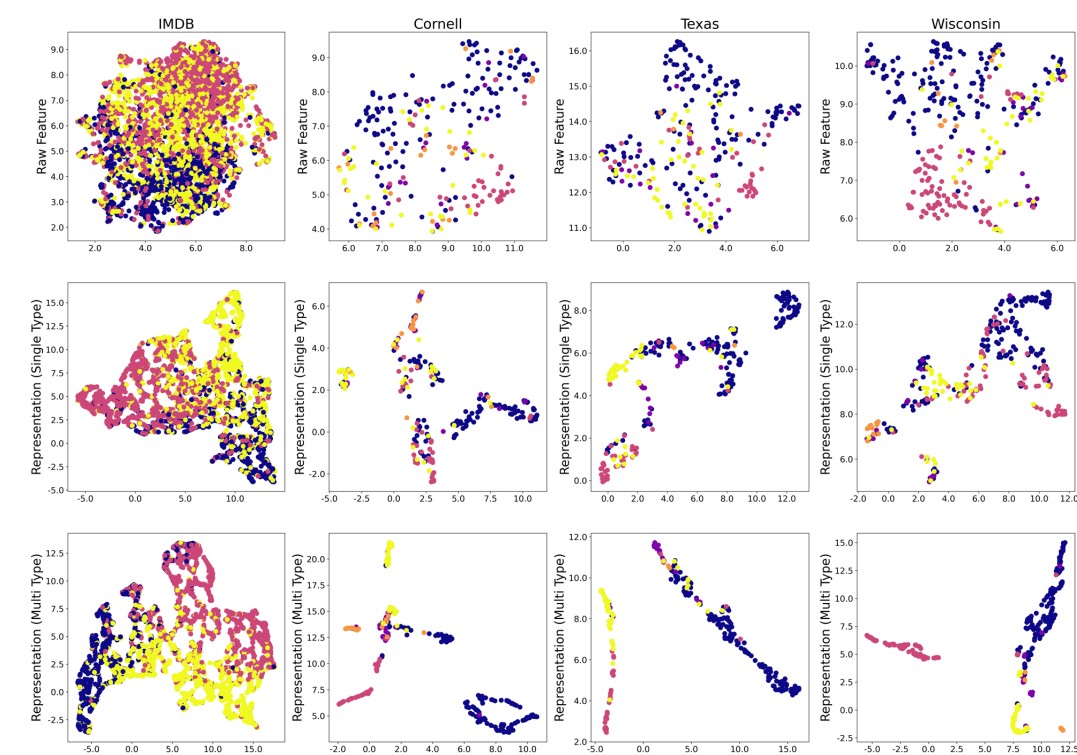

Figure 3: UMAP visualization analysis between raw features and representations of RGCN trained with single and multiple types of relations.

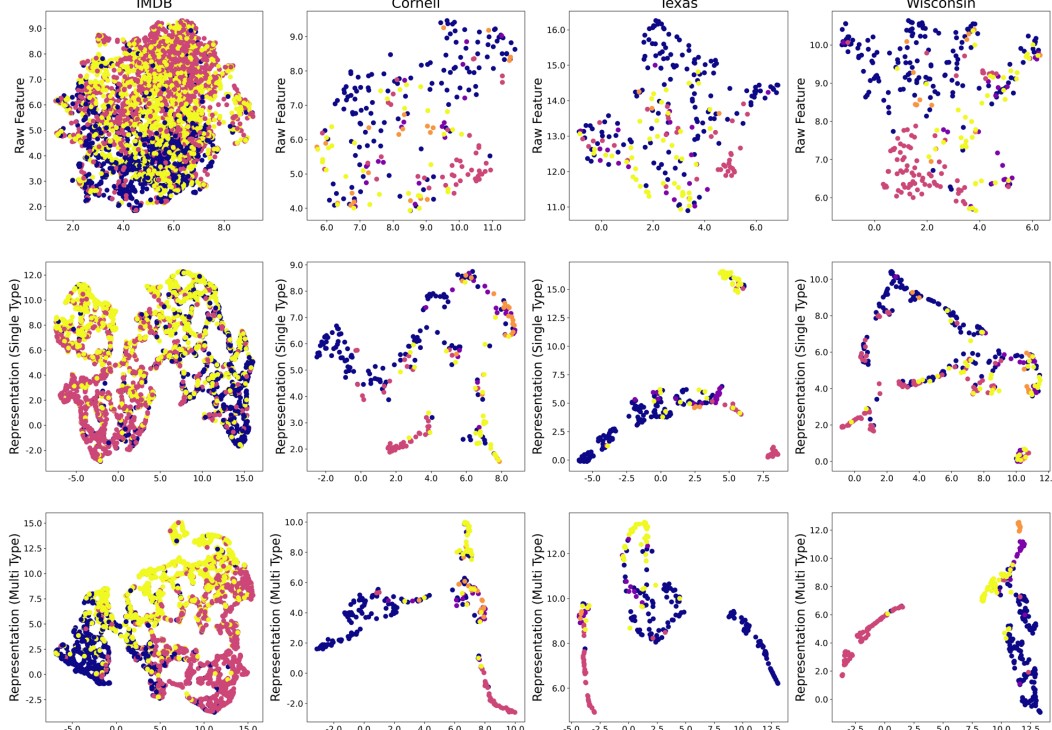

Figure 4: UMAP visualization analysis between raw features and representations of HAN trained with single and multiple types of relations.

**WikiCS** (Craven et al., 1998) is a hyperlink network in which nodes represent web pages and edges are hyperlinks. The text attribute of each node represents the web page content. The predefined categories are

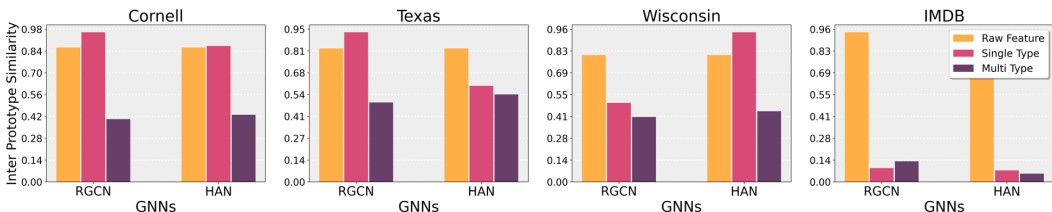

Figure 5: Comparison of average inter-prototype similarity (i.e., average cosine similarity between per-class mean representation vectors) between raw features and representations of GNNs trained with single and multiple types of relations.

Computational linguistics, Databases, Operating systems, Computer architecture, Computer security, Internet protocols, Computer file systems, Distributed computing architecture, Web technology, and Programming language topics.

**Amazon-History** (Yan et al., 2023)   is a shopping network where nodes correspond to different types of history books, and edges indicate items that are often purchased or viewed together. Each node is labeled according to its product category.

**OGBN-arXiv** (Hu et al., 2020a)   is a citation network consisting of Computer Science (CS) papers from arXiv. Nodes represent individual papers, and directed edges denote citations between them. The node labels correspond to 40 subject categories on arXiv, such as cs.AI, cs.LG, and cs.OS, assigned by the authors and arXiv moderators.

**OGBN-Products** (Hu et al., 2020a)   is a product co-purchase network on Amazon, where nodes represent Amazon products, and edges reflect products commonly bought together. The node labels are predefined and represent 47 major product categories.

Table 14: Statistics of TAG benchmark datasets.

| Dataset | Pubmed | IMDB | Cornell | Texas | Wisconsin | Cora | WikiCS | Amazon-History | OGBN-Products | OGBN-arXiv |
|---|---|---|---|---|---|---|---|---|---|---|
| #Nodes | 19,717 | 4,182 | 247 | 255 | 320 | 2,708 | 11,701 | 41,551 | 54,025 | 169,343 |
| #Edges | 44,338 | 47,789 | 213 | 119 | 449 | 5,278 | 216,123 | 358,574 | 74,420 | 1,166,243 |
| #Classes | 3 | 3 | 5 | 5 | 5 | 7 | 10 | 12 | 47 | 40 |
| Domain | Citation | Movie | Hyperlinks | Hyperlinks | Hyperlinks | Citation | Hyperlinks | Shopping | Shopping | Citation |

Comprehensive statistics of the datasets used in our experiments, including the graph domain and the number of nodes, edges, classes, are provided in Table 14.

## C.2 IMPLEMENTATION DETAILS

We adopted Sentence-BERT (Reimers & Gurevych, 2019) to encode node features and relational features when using edge-featured GNNs. To carefully identify qualified relation types, we employ Claude Opus[2] (Chat version) from Anthropic as the *relation generator* and *discriminator*. The edge decomposition is performed using a LLaMA3 (Touvron et al., 2023)-based *relation decomposer*, which is a free, open-sourced model. In our experiments, we utilize LLaMA3-8b and 70b as base LLMs, with a fixed temperature of 0.2 across all settings. Adhering to the same evaluation protocols of existing TAG works (Chen et al., 2024; He et al., 2023), we adopt the same train/validation/test splits of 60%/20%/20%, respectively. For training the GNN models, all architectures are implemented using PyTorch (Paszke et al., 2019) and PyTorch Geometric (Fey & Lenssen, 2019). All experiments are conducted on RTX Titan, RTX 3090 (24GB), A6000 GPU machines. Throughout all experiments, we set the hidden dimension to 64 and employ the Adam optimizer with a weight decay of 0. The best validation performance is selected within the following hyperparameter search space:

- Learning rate: $[0.001, 0.005, 0.05, 0.01]$
- Number of layers: $[2, 3]$
- Dropout: $[0, 0.1, 0.5, 0.8]$

---

[2]https://www.anthropic.com/claude

# D BROADER IMPACTS

Our work identifies a novel bottleneck in GNN performance for downstream tasks, specifically highlighting the oversimplification of graph structures. To address this, we introduce **RoSE**, a framework that decomposes edges to enhance the representational learning capabilities of GNNs. This shift in focus from node attributes, which dominated prior studies, to the structure itself represents a significant paradigm shift. By leveraging the general knowledge of LLMs, our approach opens new research avenues for improving graph structures. Our analysis demonstrates that **RoSE** significantly enhances classification performance of GNNs, particularly in datasets where GNNs have traditionally underperformed. Consequently, our work extends the applicability of GNN architectures to a broader spectrum of datasets, overcoming previous performance limitations and expanding their utility in various domains.

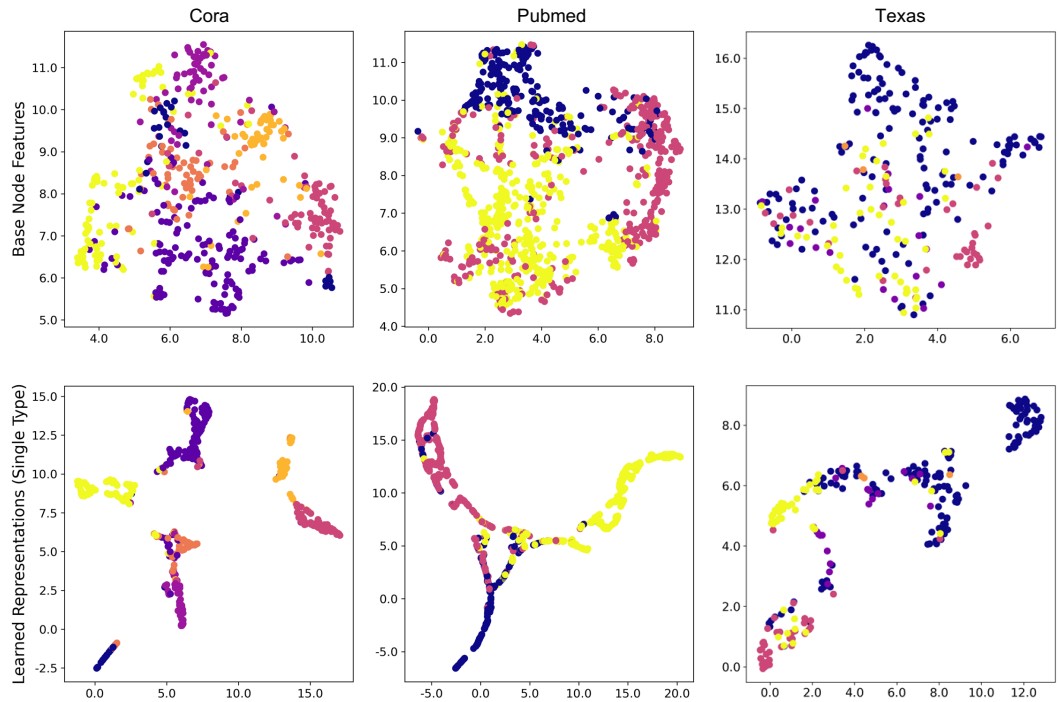

Figure 6: Comparison of raw features and learned features between Cora and Pubmed datasets versus the Texas dataset, trained with the original graph composition.

