# OpenReview forum: "Unleashing the Potential of Text-attributed Graphs: Automatic Relation Decomposition via Large Language Models"
_ICLR.cc/2025/Conference — Submitted to ICLR 2025_

### Official Review · Reviewer_XZxr · 2024-10-20

**Soundness:** 2
**Presentation:** 3
**Contribution:** 3
**Rating:** 5
**Confidence:** 4

**Summary:**

This paper proposes a prompting-based method for augmenting edge types in graphs. It contains several stages, such as relation generation, relation filtering, and relation decomposer. Experimental results on node classification show that the augmented edge types are useful.

**Strengths:**

1. Using LLM to augment the predefined graph structure is interesting.
2. The paper is well-written and easy to follow.
3. Amount of experimental results are provided to demonstrate the effectiveness of the proposed method.

**Weaknesses:**

1. My biggest concern is that the proposed method is not as totally automatic as they claimed, since for graphs from different domains, this method requires a large effort to try the best prompt for graph descriptions and relation filtering guidelines.
2. The proposed efficient relation-type annotation strategy seems too simple. The assumption that the neighbor nodes should have a similar relation type with the central node is not realistic. For example, a paper can cite one paper via a positive sentiment but cite another via a negative sentiment.
3. The proposed method can be viewed as an edge classification method. The authors should include some datasets with ground-truth edge labels to show how much the proposed method can approximate the ground-truth edge labels. In the experiment, the authors should also include the node classification performance with the ground-truth edge labels for comparison.

**Questions:**

see weaknesses

---

> ### Author Response · Authors · 2024-11-21
> **Response to Reviewer XZxr (1)**
>
> We sincerely appreciate the reviewer’s feedbacks. We would like to address the reviewer’s concerns as below.
>
> **[Streamlined Prompting of RoSE]**
>
> **W1.** We would like to first clarify that the term “automatic” in our method refers to the pipeline of our relation type identification and labeling, which eliminates the need of manual process; the determination of plausible relation types and edge decomposition are all done by LLMs.
>
> Although the prompt for these LLM agents are manually written, our prompts for relation generator and discriminator are intentionally minimal, requiring only basic and essential information. Specifically, the graph description includes a brief description of the node and edges, a sample node text attribute, and a list of predefined categories. For relation filtering, we apply two guidelines fixed across all benchmarks: (1) exclude edge types beyond the LLM's analytical scope, and (2) retain only essential types. Note that the example node text in these prompts is randomly chosen, and the number of relation types is dynamically determined by LLMs, without the need of manual supervision. By incorporating these compact information, we minimize the introduction of human bias to edge decomposition.
>
> As the reviewer pointed out, these prompts may need slight adaptation for different datasets, since individual graphs exhibit different properties (*e.g.*, citation graphs vs. co-purchasing graphs). However, once edges are semantically classified via RoSE, the enriched dataset is agnostic to the selection of GNN architecture and does not require re-processing, similar to the feature preprocessing stage. Additionally, RoSE leverages LLMs' zero-shot capabilities without needing further fine-tuning, distinguishing it from prior works [1, 2, 3, 4].
>
> ---
>
> **References**
>
> [1] Node feature extraction by self-supervised multi-scale neighborhood prediction, ICLR 2022.
>
> [2] Simteg: A frustratingly simple approach improves textual graph learning, ArXiv preprint (2023).
>
> [3] Llm-to-lm interpreter for enhanced text-attributed graph representation learning, ICLR 2024.
>
> [4] Learning on large-scale text-attributed graphs via variational inference, ICLR 2023.
>
> [5] Sentence embeddings using siamese bert-networks, EMNLP 2019.
>
> [6] Edgeformers: Graph-empowered transformers for representation learning on textual-edge networks, ICLR 2023.

---

> > ### Author Response · Authors · 2024-11-21
> > **Response to Reviewer XZxr (2)**
> >
> > **[Upon Clarification of Our Edge Decomposition]**
> >
> > **W2.** We clarify our assumptions made in RoSE-efficient. We do not assume that all neighbor nodes have a similar relation type with the central node, but rather assume a similar relation type between neighbors which have a similar feature-wise spatial relationship with the central node. Concretely speaking, for node $i$ and its neighbors $j_1$ and $j_2$, we do not assume $e_{ij_1}$ and $e_{ij_2}$ to have same relation type. Rather, we assume $e_{ij_1}$ and $e_{ij_2}$ to have same relation type if their features $z_{j_1}$ and $z_{j_2}$ are close in terms of cosine distance (Note that throughout our experiments, all node features are the embeddings from Sentence-BERT [5] PLM). Hence, in the example suggested by the reviewer, the relation types of two edges will differ as their semantics are different (positive sentiment vs. negative sentiment).
> >
> > **[On Approximation of Ground-truth Edge Types]**
> >
> > **W3.** We appreciate the reviewer’s insightful suggestion. In response, we evaluate the edge classification accuracy of our LLM-based relation decomposer and the downstream node classification performance using RGCN on the Amazon Clothing and Sports datasets, which include ground-truth edge labels. In these datasets, nodes represent users or items, while edges correspond to user reviews of items. Each edge is labeled based on the user’s rating (on a scale from 1 to 5), and item nodes are assigned sales rank group labels. Here, the node-wise text attributes are absent in these benchmarks.
> >
> > To the best of our knowledge, datasets that satisfy both (1) the existence of ground-truth edge labels and (2) suitability for LLM-based inference of these edge types (e.g., datasets where relationships are explicitly semantically defined, as opposed to co-citation or co-occurrence) are very limited. Thus, we select the Clothing and Sports datasets as the best options to demonstrate the practical potential of our approach, since they fulfill both (1) and (2). Specifically, we leveraged edge-wise text attributes (user reviews) for edge label classification. Then we initialize node features during GNN training by aggregating textual content from connected edges and embedding them using Sentence-BERT, following the practice adopted in [6].
> >
> > The results are presented in Tables below. As shown in Table 1, our relation decomposer achieves decent edge classification performance, with accuracies up to 79% on these datasets. Furthermore, using our decomposer, RoSE achieves node classification performance closely matching the use of ground-truth edge labels, with an average performance gap of only 0.34. These results demonstrate that our method effectively approximates ground-truth edge types, highlighting its practical applicability.
> >
> > Table 1. Edge classification accuracy (%) of our LLM-based decomposer (LLaMA3-8b).
> >
> > | Datasets | Clothing | Sports |
> > |---------------------------------|----------|--------|
> > | Relation decomposer         | 79.47    | 76.73  |
> >
> > Table 2. Node classification accuracy (%) of vanilla GNN, decomposition with ground-truth edge types, and RoSE.
> >
> > | Methods | Vanilla | G.T. | Ours |
> > | --- | --- | --- | --- |
> > | **Clothing** | 33.54 ± 0.31 | 37.43 ± 0.30 | 37.21 ± 0.12 |
> > | **Sports** | 33.43 ± 0.77 | 37.02 ± 0.23 | 36.57 ± 0.24 |

---

> > > ### Author Response · Authors · 2024-11-25
> > > **Looking Forward for Your Response**
> > >
> > > Dear Reviewer XZxr,
> > >
> > > Thank you once again for your valuable feedbacks.
> > >
> > > With the rebuttal period coming to a close, we kindly ask you to review our responses, as we have carefully addressed your concerns and questions.
> > >
> > > We greatly appreciate the time and effort you have dedicated to reviewing our paper.
> > >
> > > Best regards,
> > >
> > > Authors

---

> > > ### Comment · Reviewer_XZxr · 2024-11-25
> > > **Thanks for your response**
> > >
> > > Thanks to the authors for their response. My first concern has been resolved. However, I still have concerns about the model's effectiveness and am glad to have further discussion.
> > >
> > > Weakness 2: Two neighboring nodes with similar semantics may have different relationships with the central node. For instance, in a citation graph, a paper on graph learning might cite other graph learning papers (sharing similar semantic topics) either positively (e.g., "we are inspired by them") or negatively (e.g., "we reach different observations"). This challenge is particularly serious when using BERT models to encode paper semantics, as these models often struggle to capture minor semantic differences, such as inconsistency in experimental observations.
> > >
> > > Weakness 3: Thank you for providing additional results. I find them particularly interesting, as they address a very challenging task: zero-shot, training-free edge classification with an unknown number of edge categories. The proposed model demonstrates promising performance in this scenario, and I would greatly appreciate further insights into these results, which I believe will interest the graph learning community. Additionally, the user ratings involve relatively simple edge categories with limited semantics. Exploring more complex knowledge graphs, such as Wikidata, YAGO, or ICEWS—which feature node text and semantically rich edge categories—could provide a more comprehensive evaluation of the model's effectiveness.

---

> ### Author Response · Authors · 2024-11-28
> **Further Response to Reviewer XZxr (1)**
>
> We appreciate the reviewer for giving us chance to have further discussion.
>
> **[Upon Feature Encoders]**
>
> **W2.** Our RoSE-efficient is primarily designed to reduce the computational cost while preserving the performance of our original algorithm. While this focus on efficiency may entail some trade-offs in capturing intricate semantic nuances, the semantic richness of Sentence-BERT as a TAG node feature encoder has been widely acknowledged in this field, including [1, 2], with its significant performance enhancement compared to traditional node features. Furthermore, [3] evaluates the quality of generated node descriptions by measuring their embedding similarity to ground-truth descriptions, using Sentence-BERT’s encoding (referred to as the SBERT score), highlighting its effectiveness as an evaluator as well. Hence, we decide Sentence-BERT is a sufficient candidate to apply RoSE-efficient, and the effectiveness of performance preservation of RoSE-efficient with Sentence-BERT is demonstrated in our paper.
>
> Nevertheless, if the input TAG requires exceptionally high semantic fidelity, we recommend two alternative approaches: (1) using our original algorithm, RoSE or (2) employing advanced feature encoders such as fine-tuned Deberta, which is also a widely adopted node feature encoder in TAG domain [1, 4, 5]. Regarding approach (2), we provide a performance comparison of Sentence-BERT and Deberta in the table below.
>
> There are two key observations worth to be highlighted — First, incorporating Deberta into a vanilla GNN further improves performance compared to Sentence-BERT, confirming that Deberta captures richer semantic information. Second, integrating Deberta-based node feature with RoSE-efficient markedly enhances performance over RoSE-efficient + Sentence-BERT, also outperforming a strong GNN+LLM baseline - TAPE [4]. In such a case where we deploy a fine-tuned feature encoder to enhance the semantic modeling capabilities, we find RoSE-efficient to be even more helpful, where the performance gap between RoSE-efficient and vanilla is further pronounced with fine-tuned Deberta features.
>
> Table 3. Node classification accuracy according to different feature encoders, averaged over 10 runs.
>
> | **Node classification acc. (%)** | **Cora** | **Pubmed** | **WikiCS** |
> | --- | --- | --- | --- |
> | **Sentence-BERT node feat.** |  |  |  |
> | Vanilla | 88.01 ± 0.47 | 87.98 ± 0.14 | 82.02 ± 0.23 |
> | RoSE-efficient (8b) | **90.07 ± 0.29** | **90.13 ± 0.15** | **86.42 ± 0.18** |
> | **Deberta node feat.** |  |  |  |
> | Vanilla | 89.85 ± 0.36 | 89.04 ± 0.20 | 83.15 ± 0.28 |
> | RoSE-efficient (8b) | **91.92 ± 0.50** | **95.05 ± 0.17** | **90.20 ± 0.33** |
> | TAPE | 90.15 ± 0.32 | 89.42 ± 0.17 | 82.81 ± 0.24 |
>
> ---
> **References**
>
> [1] Exploring the potential of large language models (llms) in learning on graphs, KDD Explorations 2024.
>
> [2] One for all: Towards training one graph model for all classification tasks, ICLR 2024.
>
> [3] LLaGA: Large Language and Graph Assistant, ICML 2024.
>
> [4] Llm-to-lm interpreter for enhanced text-attributed graph representation learning, ICLR 2024.
>
> [5] Learning on large-scale text-attributed graphs via variational inference, ICLR 2023.

---

> ### Author Response · Authors · 2024-11-28
> **Further Response to Reviewer XZxr (2)**
>
> **[Further Experiments on Approximating Ground-truth Edge Types]**
>
> **W3.** Thank you for your positive feedback and insightful suggestions to include knowledge graphs as additional benchmarks. To evaluate the approximation of ground-truth edge types, we provide predefined relation categories in the prompt and instruct the relation decomposer to classify the edge types based on their associated text attributes. This setup allows us to assess how *accurately* the LLM-based decomposer can infer relation types given the textual context. In our experiments on the Amazon Clothing and Sports datasets, the edge-wise text attributes are user reviews and the corresponding edge types represent user ratings of purchased items. LLMs, including LLaMA family, have been well-regarded for their strong zero-shot, training-free capabilities in text analysis and sentiment interpretation [1, 2, 3]. Hence, this enables the relation decomposer to achieve a reasonable degree of edge classification accuracy, resulting in a minimal performance gap in node classification tasks compared to settings where ground-truth edge types are directly utilized.
>
> Extending our analysis, we conducted additional experiments on two knowledge graphs, Wikidata and YAGO adopted in [4], as suggested. In these experiments, the downstream task is set to link (relation) prediction, as these benchmarks lack node labels and are originally designed for link prediction. Note that the validation and test edges are not accessible during message passing phase in GNN. Hence, the decomposer is instructed to predict the relation type between two connected entities in message-passing edge set, based on their node-wise text attributes. As shown in Table 3, our relation decomposer shows decent classification performance in knowledge graphs as well, leveraging significant zero-shot capabilities of LLMs. This leads to notable performance enhancement compared to vanilla GNN (with no edge decomposition), as well as demonstrating comparable performance to the setting with ground-truth edge type-based decomposition. Furthermore, the results validate the efficacy of our proposed approach in scenarios involving semantically rich and complex edge categories.
>
> We hope these results address your remaining concerns.
>
> Table 4. Edge classification accuracy (%) of our LLM-based decomposer.
>
> | **Edge classification acc. (%)** | **Wikidata** | **YAGO** |
> | --- | --- | --- |
> | Relation decomposer (8b) | 66.31 | 82.56 |
>
> Table 5. Link prediction performance of vanilla RGCN, decomposition with ground-truth edge types, and RoSE, averaged over 10 runs (higher, the better).
>
> | **MRR (%)** | Vanilla | G.T. | Ours |
> | --- | --- | --- | --- |
> | **Wikidata** | 23.88 ± 0.90 | 38.00 ± 3.70 | 36.22 ± 1.93 |
> | **YAGO** | 38.03 ± 0.48 | 49.35 ± 0.76 | 50.00 ± 0.50 |
>
> ---
> **References**
>
> [1] LLaGA: Large Language and Graph Assistant, ICML 2024.
>
> [2] Exploring the potential of large language models (llms) in learning on graphs, KDD Explorations 2024.
>
> [3] Label-free Node Classification on Graphs with Large Language Models, ICLR 2024.
>
> [4] Learning to Sample and Aggregate: Few-shot Reasoning over Temporal Knowledge Graphs, NeurIPS 2022.

---

> ### Author Response · Authors · 2024-11-29
> **Additional Insights of Experiments on Table 1 & 2**
>
> Dear Reviewer XZxr,
>
> We kindly present additional case studies on our relation decomposer, to provide enriched insights of the impressive performance on approximating ground-truth edge types. As shown in the following examples on Amazon Clothing dataset, our decomposer can well-distinguish relatively unsatisfactory, moderate, and satisfactory sentiments (please refer to the bold-faced parts), resulting in promising edge classification accuracy and thereby significantly improving the node classification performance.
>
> **[Raw text attributes and corresponding predicted edge types (ratings) from relation decomposer]**
>
> Predicted rating: 1 (unsatisfactory)
>
> >I am between S and M depending on the brand in clothes. Read the reviews here and decided on size L for this one. It barely fits me. **It was a struggle to hook it in the front**, when I finally got it on, it is extremely tight, **I can barely breathe and certain movements will cause you pain.** I should have gotten size XL to start with, but I will keep it, as it is incredibly motivating to try and work out and lose weight and my body looks fantastic in dresses and pants and dresses that barely fit me looked fabulous on me. I can only wear it for a few hours a day. **Wearing it to work is not recommended unless you got the size right**
>
> Predicted rating: 3 (moderate)
>
> >**Delivery was perfect**. I ordered a large for my 5'3" and 150 lb frame. **My complaint is that the dress was a bit too snug across the bust and the bodice was short.** I know how this dress should fit. I grew up seeing my aunt fitting dresses like these for her clients.
>
> Predicted rating: 5 (satisfactory)
>
> >**These earrings are perfect** when you want to go big and fancy without spending a lot of money!  **I am impressed** by how “real” these earrings look.  And you don’t have to worry about losing them.  **Great price for a great look.**

---

### Official Review · Reviewer_ndzw · 2024-11-03

**Soundness:** 3
**Presentation:** 3
**Contribution:** 3
**Rating:** 6
**Confidence:** 4

**Summary:**

The paper presents a framework RoSE (Relation-oriented Semantic Edge-decomposition) to enhance the representation learning of Graph Neural Networks by decomposing edges of TAGs into semantically meaningful relations. The authors highlight the limitations of traditional TAG methods that treat all edges uniformly and demonstrate that automatic edge decomposition using LLMs significantly improves node classification performance. The framework operates in two stages: (1) identifying potential relation types with an LLM-based generator and discriminator, and (2) categorizing each edge into identified relation types using LLMs. Extensive experiments show that RoSE leads to up to a 16% improvement in node classification accuracy on benchmark datasets.

**Strengths:**

1. The introduction of RoSE as an automated method to address the challenge of edge oversimplification in TAGs is interesting.
2. The authors provide extensive experiments that demonstrate substantial improvements across multiple datasets and GNN architectures.
3. The introduction of an efficient edge-sampling strategy to minimize LLM queries is a thoughtful addition.

**Weaknesses:**

1. One of the biggest concerns is edge oversimplification in TAGs may not be a problem. There is a type of graph called textual-edge graphs, where edges are annotated with rich text information [1, 2].
2. Although RoSE introduces methods to improve efficiency, the initial decomposition process may still be resource-intensive for practitioners working with massive TAGs. A more detailed analysis of the computational cost compared to traditional methods would strengthen the paper.
3. The relation discriminator's role and effectiveness are crucial to RoSE, yet the description of its implementation and performance impact lacks clarity. Further explanation or ablation studies would help clarify its significance.

[1] Li, Zhuofeng, et al. "TEG-DB: A Comprehensive Dataset and Benchmark of Textual-Edge Graphs." arXiv preprint arXiv:2406.10310 (2024).
[2] Ling, Chen, et al. "Link Prediction on Textual Edge Graphs." arXiv preprint arXiv:2405.16606 (2024).

**Questions:**

1. Can you provide a more in-depth analysis of the computational cost associated with RoSE, especially for large-scale datasets? A direct comparison with simpler baselines could give a clearer picture of the trade-offs involved.
2. Can you elaborate on the design and function of the relation discriminator, including specific examples and results from ablation studies?

---

> ### Author Response · Authors · 2024-11-21
> **Response to Reviewer ndzw (1)**
>
> We sincerely appreciate the reviewer’s feedbacks. We would like to address the reviewer’s concerns as below.
>
> **[Upon Clarification of Our Setting]**
>
> **W1.** We thank the reviewer for the insightful comment. However, we respectively disagree with the reviewer’s perspective, for three aspects: (1) Prevalence of **graphs without such text attributed edges**, (2) Our observation on edge simplification goes beyond performance gains in current dataset but **further provides insights on future dataset curation works.**, and finally that (3) RoSE offers **flexibility** to pre-constructed datasets. We detail each assertion in the following paragraphs.
>
> **(1) Prevalence of “Conventional Graphs”**
>
> We acknowledge that some datasets [1] include textual attributes as edge features. However, we emphasize that this is not the case for many real-world graphs [2, 3, 4, 5, 6, 7, 10], where datasets are often pre-constructed without such textual edge descriptions or types. Our work specifically aims to enhance the performance of GNNs on these textual-attributed graphs that lack edge-level textual attributes during the dataset construction process.
>
> In such scenarios, two main options arise: (a) manually labeling edges with textual attributes or (b) automatically generating textual descriptions for edges. RoSE provides a superior solution compared to both approaches. It avoids the substantial cost of manual labeling by leveraging automation through LLMs while delivering significantly better performance gains than naïve methods for generating edge descriptions. The empirical results in Table 1 below validate this assertion. As shown, RoSE surpasses GNNs trained with naïve textual edge descriptions from LLMs, highlighting the empirical significance of our semantic decomposition.
>
> **(2) Insights to Future Data Curation Works**
>
> We further emphasize the significance of our work, particularly in highlighting the critical role of edge simplification for text-attributed graphs (TAGs) in the construction of future datasets. Specifically, our observation underscores the importance of ensuring *discernibility* among edges during the dataset construction process of text-attributed graphs with edge attributes. By prioritizing the semantic differentiation of edges - either in terms of feature vectors or raw text, dataset creators can facilitate more effective downstream tasks, enabling GNNs to better capture meaningful relationships within the graph.
>
> **(3) Offering flexibility to pre-constructed datasets**
>
> Additionally, we believe RoSE has the potential to provide additional flexibility to pre-constructed datasets. More specifically, the optimal edge attributes for a given TAG may differ when the task of focus changes (*e.g.*, addition / exclusion of labels of node classification, change of  classes in node classification). In such a scenario, manually re-identifying and assigning relevant edge attributes demands significant burden. However, when incorporated with RoSE, the graph's edge attribute can be dynamically changed based on the targets and tasks, where the relation generator and discriminator selects appropriate set of edge descriptions to solve the given task. This offers engineers an additional flexibility in their training process, with a high possibility (demonstrated through the versatility of our work) of performance gain.
>
> Table 1. Node classification accuracy (%) of RoSE and GNNs with textual edge descriptions, averaged over 10 runs.
>
> | LLaMA3-8b | **Methods** | **Texas** | **Cora** | **IMDB** |
> | --- | --- | --- | --- | --- |
> | GIN | LLM-based textual edge description | 71.18 ± 1.92 | 86.80 ± 0.39 | 69.50 ± 0.57 |
> |  | **RoSE** | 74.51 ± 2.13 | 88.55 ± 0.30 | 68.27 ± 0.69 |
> | UniMP | LLM-based textual edge description | 74.51 ± 1.71 | 87.75 ± 0.53 | 69.07 ± 0.73 |
> |  | **RoSE** | **76.08 ± 1.79** | **89.17 ± 0.54** | **69.55 ± 0.62** |
>
> ---
>
> **References**
>
> [1] TEG-DB: A Comprehensive Dataset and Benchmark of Textual-Edge Graphs, ArXiv preprint (2024).
>
> [2] Automating the construction of internet portals with machine learning, Information Retrieval 2000.
>
> [3] A wikipedia-based benchmark for graph neural networks, ArXiv preprint (2020).
>
> [4] Collective classification in network data. AI magazine 2008.
>
> [5]  Learning to extract symbolic knowledge from the world wide web, AAAI/IAAI 1998.
>
> [6] A comprehensive study on text-attributed graphs: Benchmarking and rethinking, NeurIPS 2023.
>
> [7]  Open graph benchmark: Datasets for machine learning on graphs, NeurIPS 2020.
>
> [8] Node feature extraction by self-supervised multi-scale neighborhood prediction, ICLR 2022.
>
> [9] Simteg: A frustratingly simple approach improves textual graph learning, ArXiv preprint (2023).
>
> [10] Llm-to-lm interpreter for enhanced text-attributed graph representation learning, ICLR 2024.
>
> [11] Learning on large-scale text-attributed graphs via variational inference, ICLR 2023.

---

> ### Author Response · Authors · 2024-11-21
> **Response to Reviewer ndzw (2)**
>
> **[Computational Cost of RoSE]**
>
> **W2 & Q1.** We appreciate the reviewer’s helpful feedback. In response, we present the comparison a detailed comparison of the time and space complexity in Table 2, on large-scale datasets with RGCN as a backbone. For comprehensive analysis, we also include the complexity analysis of GLEM [11], a recent baseline for TAGs that enhances node features through co-training between GNN and LM.
>
> Although our method incurs additional complexity (an average of 121.19 and 23.19 minutes for RoSE and RoSE-efficient, respectively) when compared against vanilla GNN, it offers an effective balance between computational efficiency and performance when compared with GLEM. As shown in the table, both RoSE and RoSE-efficient outperform GLEM. Furthermore, RoSE-efficient reduces the total runtime by an average of 96.94 minutes and peak memory usage by 20,508 MB compared to GLEM.
>
> It is also noteworthy that once the edges are semantically classified, the enriched dataset remains consistent across GNN architectures and requires no additional re-processing. Furthermore, unlike prior methods [8, 9, 10, 11], RoSE leverages language models for relation inference without requiring further fine-tuning, making it both efficient and adaptable.
>
> Table 2. Computational cost analysis of vanilla GNN, GLEM, RoSE, and RoSE-efficient. The comparison is conducted on the machine with NVIDIA RTX A6000 GPU/Intel(R) Xeon(R) Gold 6226R CPU @2.90GHz.
>
> | LLaMA3-8b |  | Vanilla | GLEM | **RoSE** | **RoSE-efficient** |
> | --- | --- | --- | --- | --- | --- |
> | **Peak Memory (MB)** | **History** | 1,276 | 41,826 | 16,504 | 16,504 |
> |  | **Products** | 1,318 | 31,656 | 15,962 | 15,962 |
> | **Duration (Total procedure)** | **History** | 7.21 sec. | 132.00 min. | 199.30 min. | 32.88 min. |
> |  | **Products** | 12.97 sec. | 108.6 min. | 43.41 min. | 13.84 min. |
> | **Accuracy** | **History** | 81.27 ± 0.13 | 84.42 ± 0.08 | 85.06 ± 0.11 | 84.87 ± 0.09 |
> |  | **Products** | 69.34 ± 0.09 | 73.96 ± 0.10 | 75.26 ± 0.17 | 74.25 ± 0.19 |
>
> **[On Contribution of Relation Discriminator]**
>
> **W3 & Q2.** In response to the reviewer’s comment, we elaborate the design and impact of our relation discriminator in RoSE.
>
> After the relation generator produces a set of plausible relation types, the relation discriminator refines this set by filtering out types that are either infeasible or redundant. The filtering process is based on two guidelines fixed across all benchmarks: (1) exclude relation types beyond the LLM's analytical scope, and (2) retain only essential relation types. Using these guidelines, along with basic graph descriptions (node/edge representations, predefined categories, and a randomly sampled text attribute), the discriminator removes noisy candidates that the generator may have initially produced. We provide detailed prompt templates for relation discriminator in Appendix A (p. 15).
>
> The effectiveness of the relation discriminator is shown in Table 4, where consistent improvements are observed across 23 out of 24 settings when the discriminator is applied, compared to using unfiltered relation types (only the generator and decomposer). Additionally, we illustrate the types retained and filtered out by the discriminator in Table 3 (main manuscript, p. 7) and Table 10 (Appendix, p. 18). In Table 3 for the Cora dataset, the generator produces plausible but overlapping types like “complementary insights” and “contrasting approaches,” as well as infeasible types such as “performance benchmark”, which is difficult to infer from nodes’ text attributes (*i.e.*, paper abstracts). The discriminator effectively removes such redundant or infeasible types, resulting in a more focused set of relation types. Similarly, in Table 10 for the Texas dataset, the “studies_under/has_student” relation is redundant considering the presence of the “advised_by/advises” relation, and therefore filtered out by the relation discriminator.
>
> Hence, these results demonstrate the relation discriminator’s role in refining the relation set, leading to performance enhancement by retaining only essential and distinct relation types.

---

> > ### Author Response · Authors · 2024-11-25
> > **Looking Forward for Your Response**
> >
> > Dear Reviewer ndzw,
> >
> > Thank you once again for your valuable feedbacks.
> >
> > With the rebuttal period coming to a close, we kindly ask you to review our responses, as we have carefully addressed your concerns and questions.
> >
> > We greatly appreciate the time and effort you have dedicated to reviewing our paper.
> >
> > Best regards,
> >
> > Authors

---

### Official Review · Reviewer_efoe · 2024-11-05

**Soundness:** 2
**Presentation:** 4
**Contribution:** 3
**Rating:** 5
**Confidence:** 4

**Summary:**

This paper focus on the study of including LLMs in enhancing the textual meaning of edge connections in the graph, and then use multi-relation types graph to improve the prediction performance compared to using single-type graph structure. The authors first use a LLM generator to generate the possible types of edges, and another LLM discriminator is adopted to choose the appropriate edge type(s). A multi-relational GNN is applied on the obtained graph for prediction targets.

**Strengths:**

1. The presentation of this paper is clear and easy to follow.
2. The overall idea of enhancing the edges with semantic level types is interesting.
3. The results from human labeled edge types have shown promising potential of adopting enriched semantic edges idea.

**Weaknesses:**

1. The efficiency is a main concern. As the proposed framework requires LLM on both edge types generation and discrimination, it requires significant amount of time in these steps for both training and inference stage. This largely harms the practical potential of this method.
2. The experimental results on different datasets varied a lot. On some datasets such as Cora and Pubmed, the improvement of proposed method is very limited. Can authors give intuitive reasons behind it?
3. In some cases the semantic description between two nodes may not be able to be classified into several types, or require some domain knowledge. How the proposed method can handle those situations?

**Questions:**

Will the number of generated types (5, 10, 20, etc) affect the results a lot?

---

> ### Author Response · Authors · 2024-11-21
> **Response to Reviewer efoe (1)**
>
> We sincerely appreciate the reviewer’s feedbacks. We would like to address the reviewer’s concerns as below.
>
> **[Efficiency of Relation Generator/Discriminator]**
>
> **W1.** To clarify, our relation generator and discriminator are prompted only once per dataset, prior to GNN training. In this initial step, the generator produces plausible relation candidates, which are subsequently filtered by the discriminator. This entire process is completed within a single round of prompts, with no need for additional prompting during GNN training or inference. Moreover, in inductive setting, only the relation type assignment is required, without the need to send new queries to the relation generator or discriminator. Unlike prior methods [1, 2, 3, 4], RoSE utilizes language models for relation inference without necessitating further fine-tuning, ensuring adaptability and ease of use.
>
> **[Performance on Cora & Pubmed Datasets]**
>
> **W2.** We appreciate the reviewer’s thoughtful comment. As discussed in Section 3, decomposing edges into multiple semantic relations enhances node representations by making them more distinguishable. In datasets like Cora and Pubmed, however, the original GNN accuracy is already high, indicating that the node features themselves are sufficiently distinguishable (please refer to the Figure 6 in Appendix of our revised manuscript). Additionally, both datasets are categorized as homophilous graphs, suggesting that GNNs achieve high representation quality through neighborhood aggregation alone. As a result, the performance enhancement when integrated with our method could be limited for further improvement. The benefits of our method is more pronounced in cases where GNNs face challenges due to their ambiguous node representations. Nevertheless, as demonstrated in the table below, our method can achieve superior performance compared to TAPE [3] (a strong recent GNN-LLM baseline), when combined with a fine-tuned feature encoder.
>
> Table 1. Node classification accuracy (%) of RoSE and TAPE integrated with Deberta feature encoder, utilizing RGCN as a backbone.
>
> |  | Cora | Pubmed |
> | --- | --- | --- |
> | TAPE (w/ Deberta feat.) | 90.15 ± 0.32 | 89.42 ± 0.17 |
> | **RoSE (w/ Deberta feat.)** | **92.47 ± 0.50** | **95.59 ± 0.16** |
>
> ---
>
> **References**
>
> [1] Node feature extraction by self-supervised multi-scale neighborhood prediction, ICLR 2022.
>
> [2] Simteg: A frustratingly simple approach improves textual graph learning, ArXiv preprint (2023).
>
> [3] Llm-to-lm interpreter for enhanced text-attributed graph representation learning, ICLR 2024.
>
> [4] Learning on large-scale text-attributed graphs via variational inference, ICLR 2023.
>
> [5] Sentence embeddings using siamese bert-networks, EMNLP 2019.

---

> > ### Author Response · Authors · 2024-11-21
> > **Response to Reviewer efoe (2)**
> >
> > **[On Applicability]**
> >
> > **W3.** Thank you for your constructive feedback. First, we clarify on the problem scenario of RoSE. Our work focuses on TAGs where edges inherently represent semantic relations, such as “contrasting approaches to a similar problem”. The discussion of cases where relationships between nodes cannot be semantically classified from node attributes is beyond our current scope.
> >
> > We also argue that, the very act of modeling into a graph structure in real-world TAGs inherently involves determining some form of semantic relations beforehand. For instance, in the OGBN-Products dataset, the edges between product nodes are semantically represented as “products are frequently co-purchased together”. Moreover, many TAGs possess inherent semantic relations from associated nodes’ contents. For example, in the WebKB datasets, the edges between web page nodes are hyperlinks, which are often created based on the specific contextual interaction between pages. Our work addresses these scenarios frequently found in real-world graphs by decomposing edges into multiple “sub-explainable relations”.
> >
> > In the scenario proposed by the reviewer, where semantic edge decomposition via LLMs cannot be utilized (*e.g.*, relationships cannot be classified into distinct types or require extreme domain knowledge), we suggest that edge decomposition based on semantic distance between node pairs can serve as an alternative.Specifically, node features can be encoded from their raw text attributes using a pretrained language model (PLM), and edges can be categorized into two groups: (a) similar edges and (b) dissimilar edges, based on the cosine similarity of these node features. In this way, the relations can be semantically decomposed in a high level.
> >
> > To validate this strategy, we present node classification results for three scenarios in Table 2: (1) Vanilla GNNs with PLM-encoded node features, lacking our edge decomposition method, (2) Distance-based decomposition using TF-IDF-encoded node features (TF-IDF distance), and (3) Distance-based decomposition using PLM-encoded node features (Semantic distance). We adopt TF-IDF encoding as one of the baselines to better isolate and evaluate the impact of semantic edge decomposition. Consistent with our quantitative analysis in main paper, we choose Sentence-BERT [5] as the PLM backbone and adopted Texas, Cora, IMDB as benchmarks.
> >
> > The results highlight a key observation: semantic edge decomposition significantly enhances the performance of GNNs. Notably, performance improvements are observed only when PLM features are used for distance calculation, as they capture semantic meaning. In contrast, using TF-IDF features, which lack semantic context, results in a performance decline. Therefore, we believe that distance-based semantic decomposition can provide a viable alternative in the scenario described by the reviewer. However, to *fully* realize the performance gains enabled by rich semantic decomposition, LLM-based approaches like RoSE are essential, as they achieve even greater improvements and deliver state-of-the-art results.
> >
> > Table 2. Node classification accuracy (%) of vanilla GNNs and different decomposition strategies, averaged over 10 runs.
> >
> > | **GNNs** | **Decomposition method** | **Texas** | **Cora** | **IMDB** |
> > | --- | --- | --- | --- | --- |
> > | **RGCN** | Vanilla | 65.88 ± 1.86 | 88.01 ± 0.47 | 62.96 ± 0.44 |
> > |  | TF-IDF distance | 61.57 ± 2.64 | 81.14 ± 0.95 | 61.17 ± 0.56 |
> > |  | **Semantic distance** | **66.67 ± 2.15** | **88.03 ± 0.46** | **66.99 ± 0.48** |
> > | **GIN** | Vanilla | 68.63 ± 1.73 | 87.05 ± 0.36 | 67.59 ± 0.41 |
> > |  | TF-IDF distance | 68.63 ± 1.47 | 70.75 ± 0.29 | 60.77 ± 0.44 |
> > |  | **Semantic distance** | **72.94 ± 1.88** | **87.94 ± 0.41** | **69.12 ± 0.68** |
> >
> > **[Determining the Number of Relation Types]**
> >
> > **Q1.** In RoSE, the number of relation types is not set as a fixed hyperparameter. The relation generator and discriminator dynamically controls the number of relations, making it difficult to predefine or adjust the number of relation types. As described in Appendix A, we prompt the relation generator to suggest as many edge types as it deems essential for decomposition. Subsequently, we instruct the relation discriminator to retain only the relevant relation types without specifying a fixed number of relations. Such design is adopted to ensure that only the necessary relation types are selected, minimizing the inclusion of noisy or redundant (overlapping) types.

---

> > > ### Author Response · Authors · 2024-11-25
> > > **Looking Forward for Your Response**
> > >
> > > Dear Reviewer efoe,
> > >
> > > Thank you once again for your valuable feedbacks.
> > >
> > > With the rebuttal period coming to a close, we kindly ask you to review our responses, as we have carefully addressed your concerns and questions.
> > >
> > > We greatly appreciate the time and effort you have dedicated to reviewing our paper.
> > >
> > > Best regards,
> > >
> > > Authors

---

### Official Review · Reviewer_g77L · 2024-11-07

**Soundness:** 2
**Presentation:** 2
**Contribution:** 2
**Rating:** 5
**Confidence:** 4

**Summary:**

The paper identifies the problem that conventional edges on text-attributed graphs hinder the representation learning process of GNNs on downstream tasks when integrated with advanced node features. To resolve the issue the edges are decomposed into semantic relations by the LLM. This has also been proposed as a data enhancement method where edge attributes are absent.
The relations are identified using an LLM-based relation generator and discriminator through prompting. To avoid labeling every edge and making a lot of LLM queries, similar relations are applied to similar semantic relations nodes with a threshold value.

**Strengths:**

The paper presents good experiments with multiple runs to prove their hypothesis. Multiple GNN architectures have been used on various datasets which adds to the fact the technique works.

**Weaknesses:**

1. I have concerns about the novelty of the ideas presented. Generating edges' relations through LLM is not new; it has been shown to be the basic strategy for combining LLMs and GNNs.
2. As asked in the "Question section" I am not sure that the time and space complexity added would be worth the accuracy increase. Presenting those numbers would make the argument stronger.

**Questions:**

What is time increment for each process? The time table only has comparison on the sampling approach vs not using the approach. What is the time takes for -
1. Assigning edge labels
2. Training with edge attributes

---

> ### Author Response · Authors · 2024-11-21
> **Response to Reviewer g77L (1)**
>
> We sincerely appreciate the reviewer’s feedbacks. We would like to address the reviewer’s concerns as below.
>
> **[On Our Original Contribution]**
>
> **W1.** We appreciate your constructive feedback. However, we respectfully disagree with the characterization of our approach as standard LLM-GNN approaches. While prior works have indeed integrate LLMs with GNNs, they primarily focus on enhancing node-wise attributes [1, 2, 3, 4, 5, 6]. However, as we have emphasized in related works section, **all these approaches treat graph edges as uniform, binary connections without capturing the nuanced, inherent semantics that many real-world TAGs contain**. Additionally, existing methods that attempt to enrich edge attributes [7, 8] rely on settings where edges **already have ground-truth** descriptive texts or predefined relation types, limiting their applicability.
>
> In contrast, RoSE addresses this limitation by leveraging analytical capabilities of LLMs to infer and assign rich semantic relations to edges in TAGs where edge descriptions and ground-truth labels are absent. This enables conventional TAGs to be enriched with meaningful relational information, consequently leading to empirical performance improvements.
>
> ---
>
> **References**
>
> [1] Textgnn: Improving text encoder via graph neural network in sponsored search, WWW 2021.
>
> [2] Node feature extraction by self-supervised multi-scale neighborhood prediction, ICLR 2022.
>
> [3] Simteg: A frustratingly simple approach improves textual graph learning, ArXiv preprint (2023).
>
> [4] Graphformers: Gnn-nested transformers for representation learning on textual graph, NeurIPS 2021.
>
> [5] Learning on large-scale text-attributed graphs via variational inference, ICLR 2023.
>
> [6] Llm-to-lm interpreter for enhanced text-attributed graph representation learning, ICLR 2024.
>
> [7] Edgeformers: Graph-empowered transformers for representation learning on textual-edge networks, ICLR 2023.
>
> [8] Learning multiplex embeddings on text-rich networks with one text encoder, ArXiv preprint (2023).

---

> ### Author Response · Authors · 2024-11-21
> **Response to Reviewer g77L (2)**
>
> **[On Complexity of RoSE]**
>
> **W2 & Q1.** We would like to clarify that the time analysis in our manuscript encompasses the process of edge sampling and decomposition process. In addition to the time analysis provided in the main paper, we present a detailed comparison of the time and space complexity for the following scenarios: (1) edge sampling with edge label assignment (Table 1), (2) training GNNs with/without edge attributes (Table 2), and (3) overall procedure, all on large-scale datasets using RGCN (Table 3). For a thorough evaluation, we also include the complexity analysis of GLEM [5], a strong baseline that enhances node features through co-training between GNN and LM (Deberta).
>
> As shown in Table 2, incorporating edge attributes into downstream GNN training sacrifices an average overhead of only 6.81 seconds compared to training a vanilla model. Furthermore, both RoSE and RoSE-efficient outperform GLEM, as demonstrated in Table 3. Notably, RoSE-efficient reduces the total runtime by an average of 96.94 minutes and peak memory usage by 20,508 MB compared to GLEM. These results highlight that our method offers a balance between computational efficiency and performance.
>
> We also emphasize that once edges are semantically classified via RoSE, the model developer can freely choose the GNN model to be utilized. Meaning that, once processed, the dataset is agnostic to the selection of GNN architecture and does not require re-processing, analogous to the feature preprocessing stage. Additionally, unlike prior works [2, 3, 5, 6], RoSE does not require further fine-tuning of language models, leveraging the significant zero-shot capabilities of LLMs to determine semantic relations based on text.
>
> Table 1. Complexity analysis of relation type assignment (and edge sampling for RoSE-efficient).
>
> | LLaMA3-8b | **Peak Memory (MB)** | **Duration (RoSE)** | **Duration (RoSE-efficient)** |
> | --- | --- | --- | --- |
> | **History** | 16504 | 199.12 min. | 32.71 min. |
> | **Products** | 15962 | 43.04 min. | 13.49 min. |
>
> Table 2. Complexity analysis of downstream GNN training.
>
> | LLaMA3-8b |  | w/o edge attributes | **w/ edge attributes (RoSE)** | **w/ edge attributes (RoSE-efficient)** |
> | --- | --- | --- | --- | --- |
> | **Peak Memory (MB)** | **History** | 1,276 | 1,622 | 1,613 |
> |  | **Products** | 1,318 | 2,418 | 2,400 |
> | **Duration** | **History** | 7.21 sec. | 11.08 sec. | 10.47 sec. |
> |  | **Products** | 12.97 sec. | 22.71 sec. | 20.98 sec. |
>
> Table 3. Complexity analysis of total procedure. The comparison is conducted on the machine with NVIDIA RTX A6000 GPU/Intel(R) Xeon(R) Gold 6226R CPU @2.90GHz.
>
> | LLaMA3-8b |  | Vanilla | GLEM | **RoSE** | **RoSE-efficient** |
> | --- | --- | --- | --- | --- | --- |
> | **Peak Memory (MB)** | **History** | 1,276 | 41,826 | 16,504 | 16,504 |
> |  | **Products** | 1,318 | 31,656 | 15,962 | 15,962 |
> | **Duration (Total procedure)** | **History** | 7.21 sec. | 132.00 min. | 199.30 min. | 32.88 min. |
> |  | **Products** | 12.97 sec. | 108.6 min. | 43.41 min. | 13.84 min. |
> | **Accuracy** | **History** | 81.27 ± 0.13 | 84.42 ± 0.08 | 85.06 ± 0.11 | 84.87 ± 0.09 |
> |  | **Products** | 69.34 ± 0.09 | 73.96 ± 0.10 | 75.26 ± 0.17 | 74.25 ± 0.19 |

---

> > ### Author Response · Authors · 2024-11-25
> > **Looking Forward for Your Response**
> >
> > Dear Reviewer g77L,
> >
> > Thank you once again for your valuable feedbacks.
> >
> > With the rebuttal period coming to a close, we kindly ask you to review our responses, as we have carefully addressed your concerns and questions.
> >
> > We greatly appreciate the time and effort you have dedicated to reviewing our paper.
> >
> > Best regards,
> >
> > Authors

---

> > > ### Comment · Reviewer_g77L · 2024-11-27
> > > **Thank you for your response**
> > >
> > > Thank you for your response. I have increased my rating by one point.

---

> > > > ### Author Response · Authors · 2024-11-28
> > > > **Response to Reviewer g77L**
> > > >
> > > > We sincerely thank the reviewer for the positive reassessment for our work! We are glad that our responses address your concerns.

---

### Author Response · Authors · 2024-11-21
**Global Response**

Dear Reviewers and AC,

We sincerely appreciate your dedicated time and insightful feedback in improving our manuscript.

As acknowledged by the reviewers, we propose a novel observation on GNNs (efoe, ndzw) - the oversimplification of edges in prior textual attributed graphs (TAGs), and the possibility of enhancing performance via semantic edge decomposition. To address this, we propose RoSE, a novel (efoe, ndzw, XZxr) framework that utilizes the capabilities of LLMs to decompose edges into semantic categories. Our method is supported through extensive analysis on multiple benchmark datasets and architectures, demonstrating both its efficacy and versatility (g77L, ndzw, XZxr).

To address your constructive comments, we have conducted the following additional experiments and discussions:

- **Detailed computational cost analysis** including peak memory and total duration, compared with LLM-based graph enhancement methods — RoSE-efficient offers a **balance** between computational efficiency and performance.
- Additional **mitigation strategy in extreme scenarios** — *e.g.*, when relationships cannot be classified into distinct types.
- Discussion upon the importance of edge oversimplification on TAGs — **RoSE gives the second chance to pre-constructed conventional graphs**.
- Additional experiments demonstrating that **RoSE effectively approximates ground-truth edge types**.

Thank you for your guidance and support.

Sincerely,

Authors.

---

### Author Response · Authors · 2024-11-25
**Gentle Reminder**

Dear Reviewers and AC,

We thank the reviewers once more for the constructive feedbacks in improving our manuscript.

As the rebuttal period is nearing to its end, we kindly request you to review our responses, as this week (~26th) marks the end of our interaction period.

We have faithfully addressed your concerns, and further highlight the main contributions of RoSE once again:

- **Prevalence of Graphs Without Text-Attributed Edges**: For many real-world graphs [1, 2, 3, 4, 5, 6, 7], datasets are often pre-constructed without such textual edge descriptions or types. RoSE is tailored to enhance GNN performance on these graphs during the dataset construction process. Note that existing LLM-GNN approaches for TAGs primarily focus on enhancing node-wise attributes, while treating edges as uniform, binary connections without identifying the inherent semantics.
- **Insights for Future Dataset Curation**: We have revealed the importance of ensuring discernibility among edges during the dataset construction in our observation. By prioritizing the semantic differentiation of edges via RoSE - either in terms of feature vectors or raw text, dataset creators can facilitate more effective downstream tasks, enabling GNNs to better capture meaningful relationships within the graph.
- **Flexibility for Pre-Constructed Datasets**: RoSE provides additional flexibility to pre-constructed datasets. In reality, the optimal edge attributes for a given TAG may differ when the task of focus changes (*e.g.*, addition / exclusion of labels of node classification, change of classes in node classification). In such a scenario, manually re-identifying and assigning relevant edge attributes demands significant burden. However, when incorporated with RoSE, the graph's edge attribute can be dynamically changed based on the targets and tasks, offering additional flexibility to engineers in their training process, with a high performance gain.

We sincerely thank you for your time and efforts in reviewing our paper, and your insightful and constructive comments.

Thank you,

Authors.

---
**References**

[1] Automating the construction of internet portals with machine learning, Information Retrieval 2000.

[2] A wikipedia-based benchmark for graph neural networks, ArXiv preprint (2020).

[3] Collective classification in network data. AI magazine 2008.

[4] Learning to extract symbolic knowledge from the world wide web, AAAI/IAAI 1998.

[5] A comprehensive study on text-attributed graphs: Benchmarking and rethinking, NeurIPS 2023.

[6]  Open graph benchmark: Datasets for machine learning on graphs, NeurIPS 2020.

[7] Llm-to-lm interpreter for enhanced text-attributed graph representation learning, ICLR 2024.

---

### Author Response · Authors · 2024-12-02
**Gentle Reminder for Final Discussion**

Dear Reviewers,

With only two days remaining in the discussion period, we eagerly await your feedback.

We have carefully addressed your concerns, and hope our responses have resolved any issues or questions.

If you have any additional comments or concerns, please don't hesitate to let us know.

Sincerely,

Authors

---

### Meta-Review · Area_Chair_YTnb · 2024-12-19

**Metareview:**

This paper discusses RoSE, a framework aimed at enhancing Graph Neural Networks' performance on text-attributed graphs by using language models to decompose edges into meaningful semantic relations. The approach focuses on improving graph structure alongside node features, integrating advanced language processing for automated semantic edge decomposition, and implementing efficient sampling to reduce computational costs.

However, the work exhibits several critical weaknesses. The computational analysis lacks sufficient depth, particularly regarding cost evaluation and resource requirements for large-scale applications. The methodology faces significant challenges, including the need for substantial manual intervention in prompt adaptation and insufficient justification for assumptions about relation type similarity among neighboring nodes. The evaluation against available ground truth edge labels is limited, and the novelty is questionable given existing approaches in the field.

The paper demonstrates significant limitations that warrant rejection. While the approach shows potential, it primarily offers incremental improvements without substantial theoretical innovation. The practical constraints are particularly problematic, including computational overhead without clear benefits and resource demands that limit large-scale application viability. The framework's automation claims are undermined by the necessary manual effort in cross-domain prompt engineering. Technical foundations show weaknesses in sampling strategy assumptions, inconsistent performance improvements, and insufficient ground truth comparisons. The evaluation lacks comprehensive analysis of computational efficiency versus performance benefits, comparisons with simpler alternatives, and explanations for varying performance across different scenarios.

**Additional Comments On Reviewer Discussion:**

The review process highlighted significant concerns regarding this research paper. The reviewers questioned the innovative aspects of using language models for edge relation generation, particularly in the context of graphs without text-attributed edges. While the authors emphasized their distinct approach compared to previous work that focused primarily on node attributes, they faced challenges in convincingly demonstrating the novelty of their method.

The implementation aspects drew substantial criticism, particularly concerning computational requirements and scalability for larger applications. This emerged as a significant weakness of the work, as the authors did not adequately address these performance-related concerns. Additionally, questions arose about the framework's automatic capabilities and the necessity for domain-specific prompt engineering, though the authors attempted to present this as an advantage in terms of adaptability.

In their response, the authors aimed to reposition their work by highlighting their focus on graphs lacking text-attributed edges, contributions to dataset curation practices, and the framework's adaptability for existing datasets. However, several crucial issues remained unresolved. The authors did not sufficiently address the computational efficiency concerns, strengthen their theoretical foundation, or provide comparisons with simpler alternative approaches.

The response, while providing some clarity about the intended scope and application of their work, fell short of addressing these fundamental limitations. This reinforces the initial evaluation that significant improvements and developments are necessary before the paper meets the acceptance criteria. Though the authors' clarifications helped illuminate their intended contributions, the core issues identified in the initial reviews remained largely unaddressed.

---

### Decision · Program_Chairs · 2025-01-22

Reject